# Targeted production of reactive oxygen species in mitochondria to overcome cancer drug resistance

Hai Wang[1,2,3], Zan Gao [4], Xuanyou Liu[5], Pranay Agarwal[1,3], Shuting Zhao[1,3], Daniel W. Conroy[6], Guang Ji [7], Jianhua Yu[2,8], Christopher P. Jaroniec [6], Zhenguo Liu[3,5], Xiongbin Lu[9], Xiaodong Li[4] & Xiaoming He [1,2,3,10]

Multidrug resistance is a major challenge to cancer chemotherapy. The multidrug resistance phenotype is associated with the overexpression of the adenosine triphosphate (ATP)-driven transmembrane efflux pumps in cancer cells. Here, we report a lipid membrane-coated silica-carbon (LSC) hybrid nanoparticle that targets mitochondria through pyruvate, to specifically produce reactive oxygen species (ROS) in mitochondria under near-infrared (NIR) laser irradiation. The ROS can oxidize the NADH into $NAD^+$ to reduce the amount of ATP available for the efflux pumps. The treatment with LSC nanoparticles and NIR laser irradiation also reduces the expression and increases the intracellular distribution of the efflux pumps. Consequently, multidrug-resistant cancer cells lose their multidrug resistance capability for at least 5 days, creating a therapeutic window for chemotherapy. Our in vivo data show that the drug-laden LSC nanoparticles in combination with NIR laser treatment can effectively inhibit the growth of multidrug-resistant tumors with no evident systemic toxicity.

[1] Department of Biomedical Engineering, The Ohio State University, Columbus, OH 43210, USA. [2] Comprehensive Cancer Center, The Ohio State University, Columbus, OH 43210, USA. [3] Davis Heart and Lung Research Institute, The Ohio State University, Columbus, OH 43210, USA. [4] Department of Mechanical and Aerospace Engineering, University of Virginia, Charlottesville, VA, USA. [5] Division of Cardiovascular Medicine, Center for Precision Medicine, University of Missouri School of Medicine, Columbia, MO 65212, USA. [6] Department of Chemistry and Biochemistry, The Ohio State University, Columbus, OH 43210, USA. [7] Institute of Digestive Diseases, Longhua Hospital, Shanghai University of Traditional Chinese Medicine, Shanghai 200032, China. [8] Division of Hematology, The Ohio State University, Columbus, OH 43210, USA. [9] Department of Medical and Molecular Genetics and Melvin and Bren Simon Cancer Center, Indiana University School of Medicine, Indianapolis, IN 46202, USA. [10] Fischell Department of Bioengineering, University of Maryland, College Park, MD 20742, USA. Correspondence and requests for materials should be addressed to X.L. (email: xl3p@virginia.edu) or to X.H. (email: shawnhe@umd.edu)

Multidrug resistance of cancer cells is a major reason for the failure of chemotherapy[1,2]. Chemotherapy can kill the drug-sensitive cancer cells, but the drug-resistant cells left behind can cause tumor recurrence (or relapse) and even cancer metastasis[3,4]. As a result of the failure to eliminate tumor relapse in many cancer patients with chemotherapy, multidrug-resistant cancer cells have attracted a great deal of attention in the field of oncology for several decades. An important advance in the understanding of cancer drug resistance is the identification of P-glycoprotein (P-gp) and other related transporters-based efflux pumps in the plasma membrane of some cancer cells. These efflux pumps could recognize and catalyze the efflux of various anticancer drugs from cancer cells[5,6]. Rational approaches to target the mechanisms of cancer drug resistance are emerging as a novel strategy to improve the clinical outcome of chemotherapy. Nanoparticles have been explored to overcome the efflux pump-mediated drug resistance by delivering a high concentration of intracellular drugs[7,8]. However, existing nanoparticles are incapable of inhibiting the function of the efflux pump directly.

The most extensively characterized efflux pumps, including ABCB1 (known as MDR1 or P-glycoprotein), ABCC1 (known as MRP1), and ABCG2 (known as BCRP or MXR), belong to the adenosine triphosphate (ATP)-binding cassette (ABC) transporter superfamily[9,10]. Therefore, inhibiting the function of ABC transporters should be an effective strategy to overcome cancer drug resistance. As ATP is indispensable for the efflux pumps/transporters to function, cancer drug resistance may be overcome by inhibiting the production of ATP in cancer cells[11]. ATP is synthesized by ATP synthase powered by a concentration gradient of protons in mitochondria[12]. The proton gradient is generated by an electron transport chain with electrons donated mainly from nicotinamide adenine dinucleotide with hydrogen (NADH)[13]. The nature of the electron transport chain is rooted in several oxidation–reduction reactions[14]. Therefore, one possible way to inhibit the production of ATP is to consume NADH by oxidizing it into $NAD^+$.

As illustrated in Fig. 1a, here we report a lipid membrane-coated silica-carbon (LSC) hybrid nanoparticle for targeted

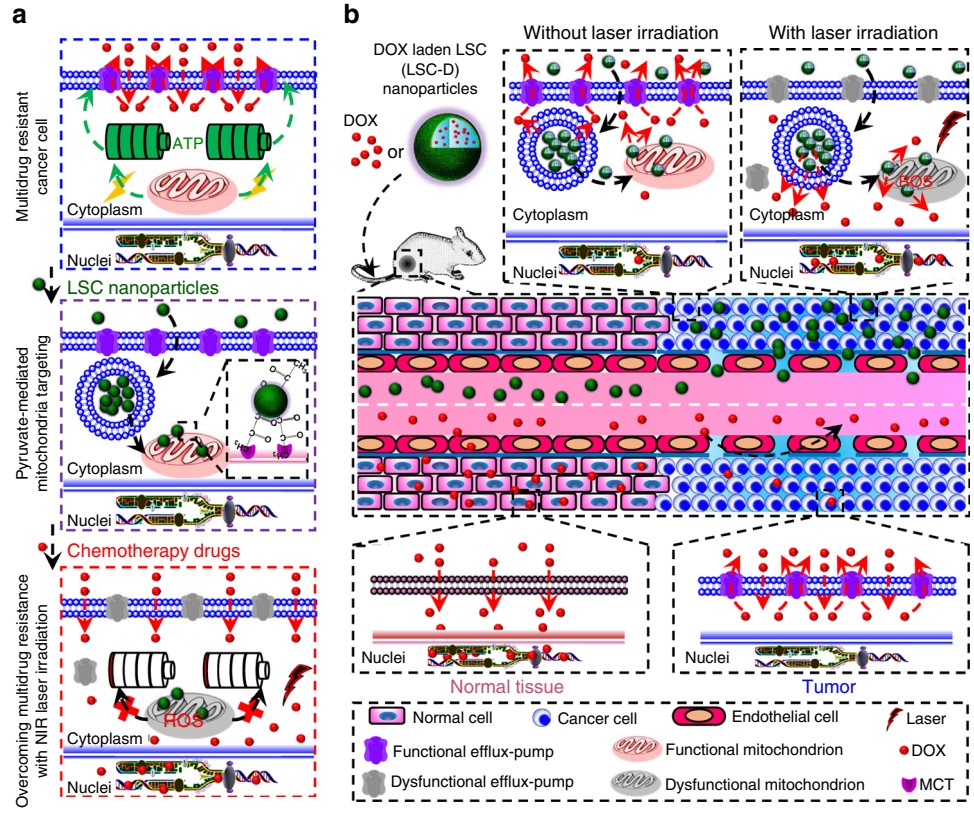

**Fig. 1** A schematic illustration of the strategy for overcoming cancer drug resistance. **a** The transmembrane P-glycoprotein (P-gp) efflux pump driven by adenosine triphosphate (ATP) is a major mechanism of cancer multidrug resistance. In this study, a novel lipid membrane-coated silica-carbon (LSC) nanoparticle is designed to target the monocarboxylate transporters (MCTs) on mitochondria through the pyruvate groups on the surface of the LSC nanoparticle. With near infrared (NIR, 800 nm) laser irradiation, the LSC nanoparticles produce reactive oxygen species (ROS) to oxidize NADH into $NAD^+$ in mitochondria, which compromises ATP production. This results in dysfunction of the P-gp efflux pumps. In addition, the treatment of LSC nanoparticles with NIR laser irradiation (LSC + L) leads to not only transmembrane but also intracellular distribution of the P-gp efflux pumps, although the total amount of the efflux pumps is only slightly (albeit significantly) reduced. Collectively, the LSC + L treatment can be used to overcome the multidrug resistance of cancer cells, which is demonstrated using three different chemotherapy drugs in this study: doxorubicin hydrochloride (DOX), paclitaxel (PTX), and irinotecan (CPT-11). **b** Owing to their nanoscale size (~45 nm), the DOX-laden LSC (LSC-D) nanoparticles can preferentially accumulate in tumor as a result of the enhanced permeability and retention (EPR) effect of tumor but not normal vasculature to minimize systemic toxicity of the chemotherapy drug. After arriving in tumor, the LSC-D nanoparticles can be taken up by multidrug-resistant cancer cells, but the DOX released out of the nanoparticles can still be pumped out of the cells for the LSC-D treatment alone. Importantly, with NIR laser irradiation, the drug resistance of the multidrug-resistant cancer cells can be overcome and effective tumor destruction ensues. In contrast, free DOX can diffuse into both normal tissue (which causes systemic toxicity) and the multidrug-resistant tumor. Moreover, free DOX can be pumped out of the multidrug-resistant cancer cells and further diffuse back into blood perfusion, leading to ineffective cancer therapy. All mouse and mitochondrion images were created using ChemBioDraw Ultra 12.0 and their color was further adjusted in Powerpoint

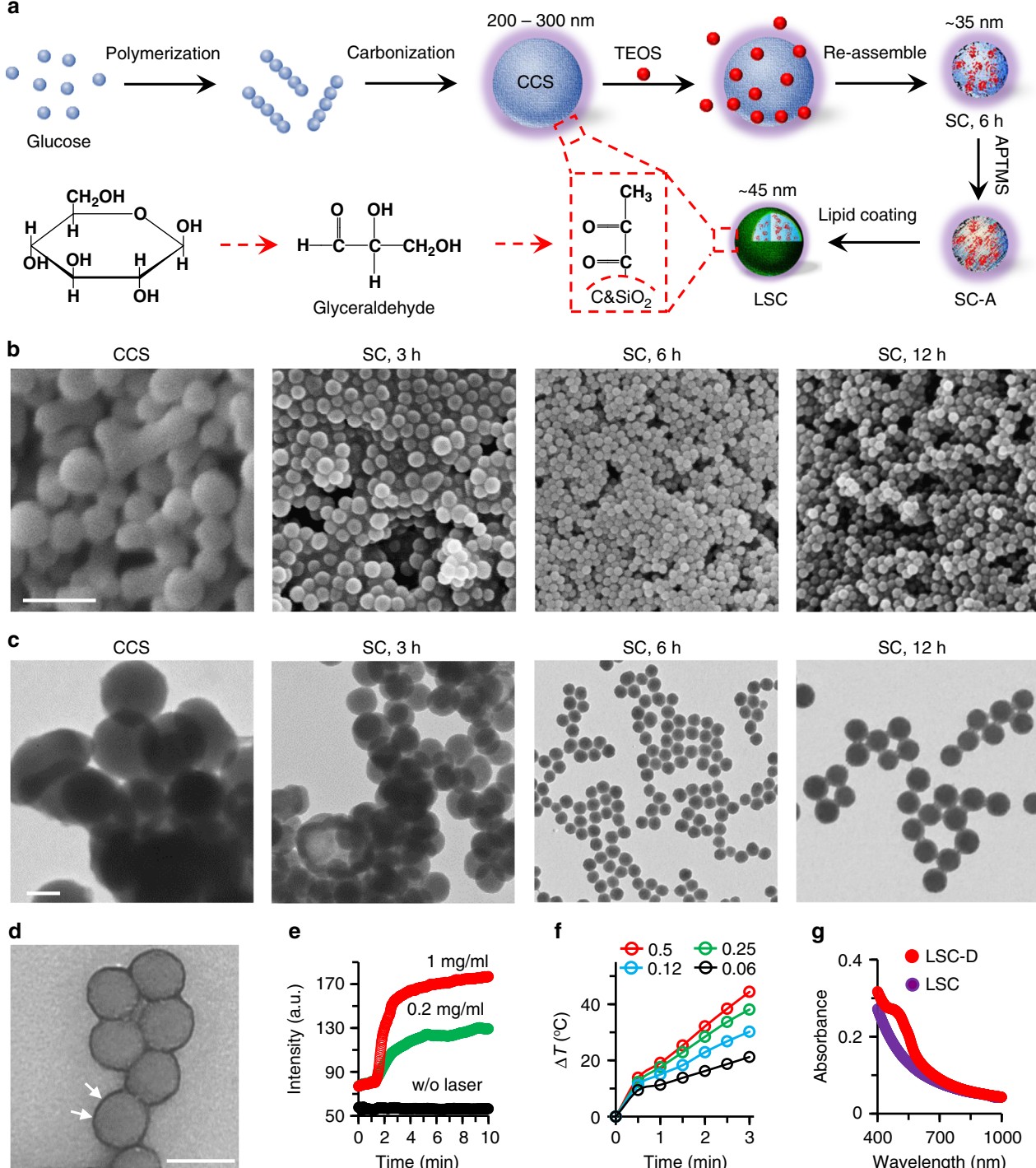

**Fig. 2** Synthesis and characterization of nanoparticles. **a** A schematic illustration of the procedure for preparing colloidal carbon sphere (CCS) nanoparticles using glucose, producing silica-carbon (SC) nanoparticles using the CCS and tetraethyl orthosilicate (TEOS), modifying the SC nanoparticles with (3-Aminopropyl) trimethoxysilane (APTMS) to form SC-A nanoparticles, and coating the SC-A nanoparticles with phospholipid dipalmitoylphosphatidylcholine (DPPC) to produce LSC nanoparticles. The pyruvate group can be formed during the hydrothermal process. **b**, **c** Scanning (SEM, **b**) and transmission (TEM, **c**) electron microscopy images of the CCS and SC nanoparticles synthesized with different reaction times showing that SC nanoparticles of ~35 nm with a homogeneous size distribution can be obtained after 6 h of reaction. Since the SC nanoparticles do not have core–shell structure, the silica should not cover the whole surface of the nanoparticles. Therefore, the lipid membrane should not completely cover the surface of the LSC nanoparticles because the lipid only interacts with silica through APTMS. Scale bar: 500 nm in **b** and 100 nm in **c**. **d** TEM image of lipid membrane-coated SC (LSC) nanoparticles showing the lipid membrane on the surface of the nanoparticles (indicated by arrows). Scale bar: 50 nm. **e** The production of singlet oxygen by the LSC nanoparticles under NIR irradiation as compared with LSC nanoparticles without NIR laser irradiation (w/o laser, 1 mg ml$^{-1}$). **f** The NIR irradiation (1.0 W cm$^{-2}$) time and LSC nanoparticle concentration (0.06–0.5 mg ml$^{-1}$) dependent increase of temperature. **g** UV-Vis absorbance of LSC nanoparticles and DOX-laden LSC (LSC-D) nanoparticles showing successful encapsulation of DOX in the LSC nanoparticles with an absorbance peak at ~480 nm

production of reactive oxygen species (ROS) in the mitochondria of tumor cells under near infrared (NIR, 800 nm) laser irradiation, which can oxidize NADH to minimize the production of ATP and overcome multidrug resistance. The combined treatment of the LSC nanoparticles and NIR laser irradiation can also decrease

the amount of the transmembrane P-gp efflux pumps by reducing the overall expression of the pump and increasing its intracellular distribution, to sensitize multidrug-resistant cancer cells to chemotherapy. Moreover, the size of the LSC nanoparticles is appropriate for utilizing the enhanced permeability and retention (EPR) effect of the tumor vasculature for in vivo tumor targeting to reduce the systemic toxicity of chemotherapy (Fig. 1b). Furthermore, unlike free drug that may be pumped out of the multidrug-resistant cancer cells to further diffuse out of tumor and back into blood perfusion, the drug released out of the LSC nanoparticles can stay in cancer cells and tumor because the function of drug efflux pumps is compromised for at least five days after NIR laser irradiation in the presence of the nanoparticles.

## Results

**Preparation and characterization of nanoparticles.** First, colloidal carbon sphere (CCS) nanoparticles were synthesized through a hydrothermal synthesis approach by using glucose (Fig. 2a)[15]. According to previous studies[16–18], pyruvaldehyde groups can be formed during the hydrothermal process (Fig. 2a). The structure of pyruvaldehyde is largely similar to pyruvate, which can specifically bind with the monocarboxylate transporters (MCTs) on the outer surface of mitochondria[19,20]. The function of MCTs is to actively transport pyruvate into mitochondria[21]. Therefore, we hypothesize that the pyruvate-decorated nanoparticles can specifically target mitochondria. Here, we used a specific pyruvate detection assay to determine its existence in CCS nanoparticles. As shown in Supplementary Fig. 1, 50 µg of CCS nanoparticles contain ~0.15 µg of pyruvate groups. However, the CCS nanoparticles are difficult to disperse in water and tend to form large aggregates (Supplementary Fig. 2a and b). According to dynamic light scattering (DLS, Supplementary Fig. 2b) and scanning (SEM, Fig. 2b and Supplementary Fig. 3) and transmission (TEM, Fig. 2c) electron microscopy analyses, the CCS nanoparticles are ~200–300 nm in diameter. This size is too large to utilize the EPR effect of tumor vasculature to passively target tumors in vivo[22–24]. In order to overcome this problem, the CCS nanoparticles were exposed to tetraethyl orthosilicate (TEOS, Fig. 2a) for varying lengths of time (3, 6, and 12 h). After 3 h of exposure, the size of the hybrid silica-CCS (SC) nanoparticles was reduced (Fig. 2b), but aggregates were still observed in SEM and TEM images (Fig. 2b, c and Supplementary Fig. 3). SC nanoparticles of ~35 nm in diameters that could be stably dispersed in water were obtained after 6 h of TEOS exposure (Fig. 2b, c, Supplementary Figs. 2a–c and 3). However, the size of the resultant SC nanoparticles increased slightly as the exposure time was extended to 12 h (Fig. 2b, c and Supplementary Figs. 2d and 3). Therefore, SC nanoparticles obtained after 6 h of exposure to TEOS were used for all subsequent experiments. The formation of SC nanoparticles with a greatly reduced size may be owing to the chemical reaction between TEOS and CCS. This is confirmed by using proton nuclear magnetic resonance ($^1$H NMR) and Fourier transform infrared (FTIR) spectroscopy (Supplementary Figs. 4–5 and Supplementary Note 1).

In order to improve the biocompatibility of the SC nanoparticles, we further modified their surface with APTMS for absorbing dipalmitoylphosphatidylcholine (DPPC), to form lipid membrane-coated SC (LSC) nanoparticles[25]. The lipid membrane

is visible in the TEM images of the resultant LSC nanoparticles (Fig. 2d). The LSC nanoparticles are ~45 nm in diameter, and can be homogeneously dispersed in water with excellent stability (Supplementary Fig. 6). Although the total amount (~0.058 µg) of pyruvate groups in 50 µg of LSC nanoparticles is reduced compared with CCS nanoparticles (Supplementary Fig. 1), it should not decrease the density of the pyruvate group on the LSC nanoparticles (Supplementary Note 2). Moreover, the LSC nanoparticles exhibit strong optical absorption in the NIR region and the absorbance at 800 nm shows a linear correlation with the nanoparticle concentration in water (Supplementary Fig. 7).

To determine the potential application of the LSC nanoparticles to produce ROS inside cells, we further identified the types of free radicals that can be produced by the nanoparticles under NIR laser irradiation (1 W cm$^{-2}$, LSC + L). A singlet oxygen ($^1O_2$) sensor was applied to specifically detect the $^1O_2$ free radical. As shown in Fig. 2e and Supplementary Fig. 8, the LSC nanoparticles exhibit an excellent photodynamic capability as demonstrated by the time- and concentration-dependent production of $^1O_2$ in deionized water. Furthermore, 5-(diethoxyphosphoryl)−5-methyl-1-pyrroline-N-oxide (DEPMPO, a spin trap for hydroxyl radicals (·OH)[26]) and terephthalic acid (TA) assays were used to confirm the production of ·OH by LSC + L with the electron paramagnetic resonance (EPR) and fluorescence spectroscopy, respectively (Supplementary Fig. 9 and Supplementary Note 3). In addition, the LSC nanoparticles have excellent photothermal effect with NIR laser irradiation (1 W cm$^{-2}$, Fig. 2f), and this photothermal effect is stable under multiple cycles (Supplementary Fig. 10).

Moreover, the LSC nanoparticles can be used for drug encapsulation and delivery owing to the silica. We encapsulated doxorubicin hydrochloride (DOX) into the LSC nanoparticles to form LSC-D nanoparticles, with an encapsulation efficiency of 82.7 ± 1.5 at a feeding ratio of 1:20 (DOX:nanoparticles). The UV-Vis absorbance spectrum of LSC-D nanoparticles has a peak at ~480 nm for DOX, which is absent from the spectrum of LSC nanoparticles, indicating successful encapsulation of DOX (Fig. 2g). The LSC-D nanoparticles are also stable in water (Supplementary Fig. 6b). The in vitro release of DOX from LSC-D nanoparticles is negligible (<4%) after 30 h of incubation at 37 °C (Supplementary Fig. 11a). Interestingly, the drug release can be triggered and precisely controlled with the dose of by NIR laser irradiation (Supplementary Fig. 11), which should be ascribed to laser induced heating[25].

**Intracellular distribution of LSC-D nanoparticles.** NCI/RES-ADR multidrug-resistant cancer cells were used in this study[27], and their resistance to free DOX was confirmed by incubating with different concentrations of free DOX. Indeed, no red fluorescence of DOX is observable in the 2D-cultured NCI/RES-ADR cells after 6 or 9 h of incubation (Supplementary Fig. 12). As controls, free DOX can enter both OVCAR-8 and MCF-7 cells, concentrate in their nuclei, and induce cytotoxicity. In contrast, red fluorescence of DOX is observable in all three types of cells when they are incubated with LSC-D nanoparticles (Supplementary Figs. 12–14). However, DOX is predominantly distributed in the cytosol and it is barely observable in the nuclei of NCI/RES-ADR cells even after 9 h of incubation (Supplementary Fig. 12). On the contrary, in OVCAR-8 and MCF-7 cells, DOX is observable in their nuclei (Supplementary Figs. 13–14). This difference is probably a result of the capability of the NCI/RES-ADR cells to pump out any free DOX slowly released out of the LSC-D nanoparticles.

Cancer stem-like cells (CSCs) are highly tumorigenic and drug resistant[28–31]. The CSCs are likely to be responsible for cancer

metastasis and tumor recurrence or relapse associated with conventional therapies[32]. Therefore, we enriched NCI/RES-ADR, MCF-7, and OVCAR-8 CSCs in their spheres by using a well-established method[33–36]. Free DOX still cannot enter the cells in the NCI/RES-ADR spheres, whereas LSC-D nanoparticles can deliver DOX into the cytosol but not the nuclei of the cells, which is similar to the DOX distribution in 2D NCI/RES-ADR cell cultures (Supplementary Fig. 15). Although DOX can enter cells in OVCAR-8 and MCF-7 spheres, the fluorescence intensity is weaker than that in the corresponding 2D-cultured cells

(Supplementary Figs. 16–17 for spheres, Supplementary Figs. 13–14 for 2D cell cultures). For OVCAR-8 and MCF-7 spheres treated with LSC-D nanoparticles, the DOX fluorescence in the cell nuclei is also weaker than that in the corresponding 2D-cultured cells. Taken together, these data suggest that using nanoparticles alone without inhibiting the efflux pump activity is inefficient for overcoming cancer drug resistance.

**Production of ROS in mitochondria to combat drug resistance.** ATP is synthesized by using the concentration gradient of

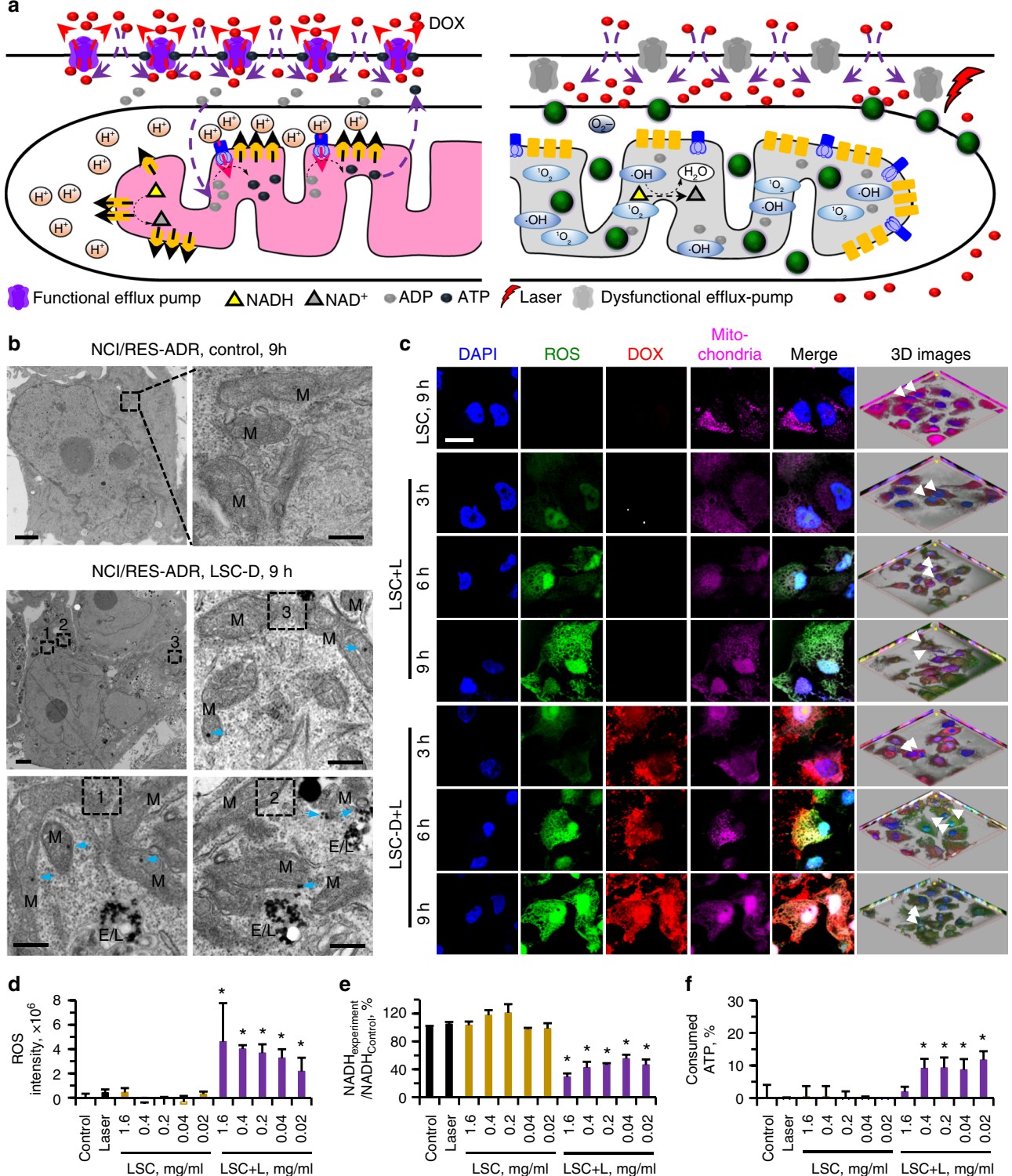

protons to power the ATP synthase located on the inner membrane of mitochondria (Fig. 3a). To maintain a higher concentration of protons outside the inner membrane of mitochondria, NADH is oxidized into $NAD^+$ to release the protons out of the inner membrane through an electron transport chain[12]. Therefore, we hypothesized that if LSC nanoparticles could target mitochondria and generate ROS in them, the ROS could oxidize NADH into $NAD^+$ directly. This should minimize the concentration gradient of protons across the inner membrane of mitochondria. As a result, ATP cannot be synthesized by ATP synthase and the efflux pump should become dysfunctional due to the lack of ATP/energy supply (Fig. 3a).

First, we checked the mitochondria-targeting capability of the LSC-D nanoparticles. After incubating 2D-cultured NCI/RES-ADR cells with the nanoparticles for 3 and 9 h, co-localization of the red fluorescence of DOX with mitochondria (green) is observable, suggesting that the LSC nanoparticles indeed can target mitochondria (Supplementary Fig. 18). To further confirm this, we checked the intracellular distribution of LSC-D nanoparticles after 9 h of incubation using biological-TEM (Bio-TEM). As shown in Fig. 3b, most of the mitochondria contain LSC nanoparticles (arrows). Some of the LSC-D nanoparticles are in the endosome/lysosome, indicating the cells take up the nanoparticles via endocytosis (Fig. 3b). Similarly, some of the red fluorescence of DOX is co-localized with endo/lysosome (green) and some is co-localized with mitochondria (purple) when both of them are labeled for confocal imaging (Supplementary Fig. 19). In contrast, lipid membrane-coated silica (LS) nanoparticles do not target mitochondria according to both the confocal and TEM images (Supplementary Figs. 20–21 and Supplementary Note 4). To further confirm the pyruvate-mediated targeting of mitochondria, we pre-treated/blocked the NCI/RES-ADR cells with pyruvic acid for 6 h before incubating them with the LSC nanoparticles. This minimizes mitochondria binding of the LSC nanoparticles (Supplementary Fig. 21b). More than 40% of the endosome/lysosome-escaped LSC-D nanoparticles are within mitochondria in the absence of pre-blocking, whereas it is 0% for the LS-D nanoparticles. With pre-blocking using pyruvic acid, the percentage decreases from >40% to ~3% (Supplementary Fig. 21c and Supplementary Note 5). It is worth noting that free DOX does not change the cellular structure (Supplementary Fig. 22 and Supplementary Note 6). As the LSC-D nanoparticles are coated with DPPC on their surface, it is interesting to understand the fate of the lipid membrane coating after cellular uptake of the nanoparticles. To this end, fluorescein isothiocyanate (FITC)-labeled DPPC (FITC-DPPC) is used to form the membrane coating on the LSC nanoparticles. As shown in Supplementary Fig. 23, the DPPC membrane can detach from the nanoparticles after cell uptake although it is stable on

nanoparticles before cell uptake (Supplementary Fig. 24 and Supplementary Note 7). Taken together, these data support that the pyruvate group on the surface of the LSC nanoparticles is responsible for their capability of targeting mitochondria.

We then investigated if LSC nanoparticles can induce the production of ROS in mitochondria. The production of ROS can be detected at 3 h after LSC + L, which further increases with time (Fig. 3c). Importantly, the ROS is well co-localized with mitochondria (note: the mitochondria tracker can stain the nuclei only after laser irradiation). Similarly, LSC-D + L-treated cells also produce strong ROS in their mitochondria. In addition, the co-localization of DOX and mitochondria confirms the mitochondria-targeting effect of the LSC-D nanoparticles (Fig. 3c). More importantly, strong DOX fluorescence can be clearly observed in the cell nuclei, indicating that the drug resistance capability of the cells is compromised after the LSC + L treatment (Fig. 3c). In order to confirm the co-localization of the ROS and mitochondria throughout the cells, 3D images were taken and ROS is observable in mitochondria throughout the cells (Fig. 3c and Supplementary Fig. 25). The quantitative data of ROS indicate that the NIR laser treatment activates the production of ROS in LSC nanoparticle-treated cells (Fig. 3d). In contrast, the ROS production by the free DOX or NIR laser irradiation treatment alone is minimal (Supplementary Fig. 26).

We further checked the amount of NADH in the cells with the various treatments. The NADH in cells treated with either LSC nanoparticles or the NIR laser alone is not significantly different from that of control (medium) group (Fig. 3e). Importantly, NADH is significantly decreased in LSC + L-treated cells. As ROS can react with NADH but not $NAD^+$, the sum of NADH and $NAD^+$ is similar for all the groups except the laser-irradiated group with the highest concentration of LSC (Supplementary Fig. 27 and Supplementary Note 8). This decrease in NADH but not the sum of NADH and $NAD^+$ is also observed in LSC + L-treated OVCAR-8 cells (Supplementary Fig. 28). To further study the function of mitochondria, a membrane-permeant dye (JC-1) was used to monitor the mitochondrial membrane potential. As shown in Supplementary Figs. 29–30, a low mitochondrial membrane potential can be detected in the LSC + L-treated cells (Supplementary Note 9). The damage to the membrane of mitochondria is further confirmed by using Bio-TEM. As shown in Supplementary Fig. 31, the membrane of mitochondria is not integral at the locations (arrows) with LSC-D nanoparticles after laser irradiation. These data indicate mitochondrial damage in NCI/RES-ADR cells treated with LSC nanoparticles and NIR irradiation, which is crucial for reducing ATP production and compromising the function of efflux pumps in the cells.

As all the efflux pumps are located across the cell plasma membrane, we isolated the membrane proteins of NCI/RES-ADR

**Fig. 3** Overcoming drug resistance by targeted production of reactive oxygen species in mitochondria. **a** When irradiated with NIR laser, the LSC nanoparticles can specifically produce reactive oxygen species (ROS) in mitochondria. This oxidizes NADH into $NAD^+$ to inhibit the production of ATP, which stops the function of the efflux pumps and overcomes the cancer drug resistance. **b** Representative TEM images of NCI/RES-ADR multidrug-resistant cells incubated in medium without (Control) or with LSC-D nanoparticles. M indicates the mitochondria. Arrows indicate the LSC-D nanoparticles in mitochondria. E/L indicates endosome/lysosome. Scale bar: 2 and 0.3 μm for low and high magnification images, respectively. **c** Confocal images of NCI/RES-ADR cells treated with LSC and LSC-D nanoparticles showing the specific production of ROS in mitochondria. Scale bar: 10 μm. **d** Quantitative data on ROS production in NCI/RES-ADR cells under various conditions. Error bars represent s.d. ($n = 3$). The LSC + L group is compared with the LSC group with the same LSC concentration, as well as the control (medium) and laser (alone) groups. $*p < 0.05$ (Kruskal–Wallis $H$-test). **e** Relative NADH in NCI/RES-ADR cells showing the decrease of NADH in cells treated with LSC nanoparticles with laser irradiation (LSC + L, 1 W cm$^{-2}$ for 1 min). Error bars represent s.d. ($n = 3$). The LSC + L group is compared with the LSC group with the same LSC concentration, as well as the control (medium) and laser (alone) groups. $*p < 0.05$ (Kruskal–Wallis $H$-test). **f** Consumption of ATP by the membrane proteins isolated from NCI/RES-ADR cells showing the transmembrane efflux pumps are not bound with ATP in the cells treated with LSC + L so that significantly more ATP can be consumed/bound by them. Error bars represent s.d. ($n = 3$). The LSC + L group is compared with the LSC group with the same LSC concentration, as well as the control (medium) and laser (alone) groups. $*p < 0.05$ (Kruskal–Wallis $H$-test)

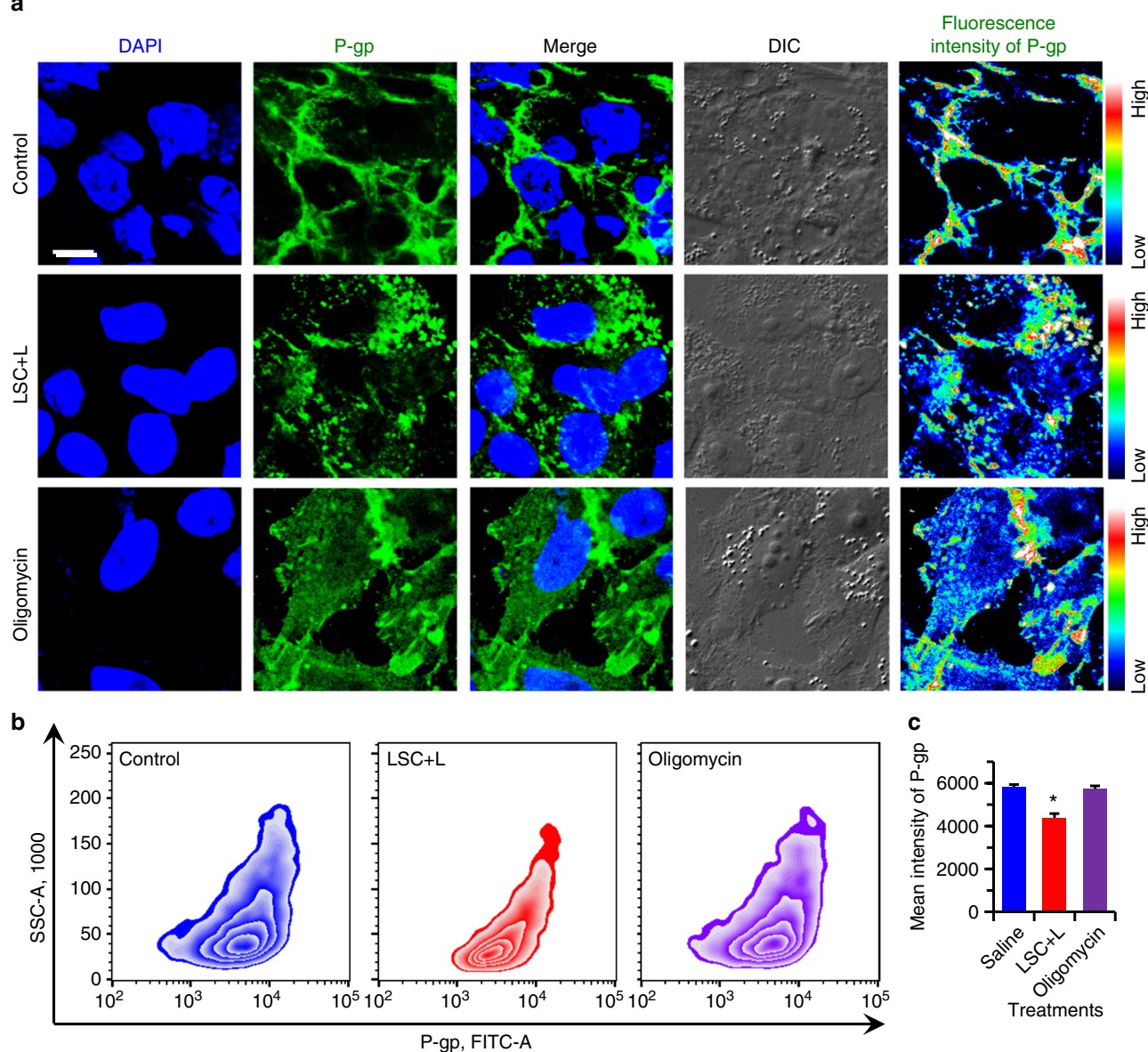

**Fig. 4** Overcoming drug resistance by reducing the amount and changing the distribution of P-gp. **a** Confocal images and fluorescence intensity of P-gp in NCI/RES-ADR multidrug-resistant cancer cells without any treatment (control) and after treated with LSC nanoparticles with laser irradiation (LSC + L, 1 W cm$^{-2}$ for 1 min). The data show that the P-gp distributes not only on the cell plasma membrane but also in the cytoplasm after the LSC + L treatment, whereas it is dominantly located on the plasma membrane for the cells in the control group. In order to confirm the changed distribution of P-gp is owing to the inhibition of ATP, the multidrug-resistant cancer cells were treated with oligomycin (an inhibitor of ATP synthase) for 12 h at 200 ng ml$^{-1}$, the results confirm that inhibition of ATP can influence the distribution of P-gp. Scale bar: 10 μm. **b–c** Flow cytometry analysis of P-gp and quantitative data show that the expression of P-gp is slightly but significantly decreased in the multidrug-resistant cancer cells with the LSC + L treatment. These data indicate that multiple mechanisms contribute to the capability of the LSC + L treatment in overcoming the drug resistance of the multidrug-resistant cancer cells. The cells were permeabilized for the immunostaining and flow cytometry analyses. Error bars represent s.d. ($n = 3$). *$p < 0.05$ (Kruskal–Wallis $H$-test)

cells after treatment with LSC nanoparticles. The isolated membrane proteins were mixed with ATP to determine whether the efflux pumps were active. As shown in Fig. 3f, the membrane proteins from cells treated with either LSC nanoparticles or NIR irradiation alone do not consume any ATP. In contrast, ATP is consumed by the membrane proteins from LSC + L-treated cells. This indicates the efflux pumps in LSC + L-treated cells are not bound with ATP. If the efflux pumps in NCI/RES-ADR cells are inactive, free DOX should be able to enter the cells and bind with the cell nuclei. Indeed, DOX is observed in both 2D-cultured NCI/RES-ADR cells and their spheres after LSC + L treatment (Supplementary Fig. 32), confirming that efflux pumps become

dysfunctional. In order to understand how long the LSC nanoparticle and laser treatment can inhibit the function of the efflux pumps, we incubated NCI/RES-ADR cells with free DOX for 3 h at 1–5 days after LSC + L treatment. As shown in Supplementary Fig. 33, free DOX can enter the cells even after 5 days although the fluorescence intensity of DOX in the cells gradually decreases with longer waiting time. These data indicate that the multidrug resistance of NCI/RES-ADR cells can be successfully overcome by targeted production of ROS in their mitochondria using a combination of LSC nanoparticles and NIR laser irradiation. This opens a therapeutic window of at least 5 days for chemotherapy.

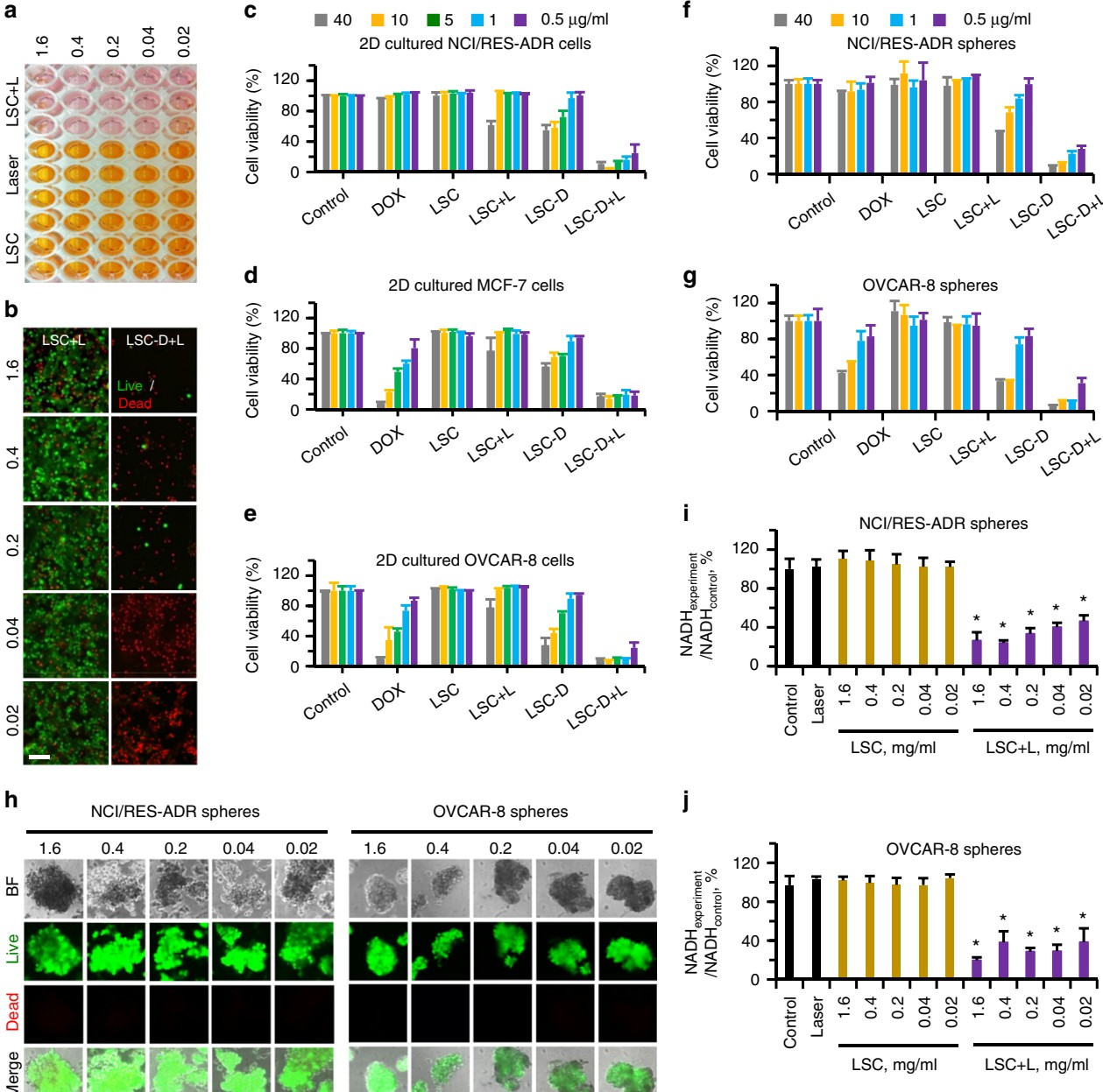

**Fig. 5** Enhanced anticancer capacity of LSC-D nanoparticles under NIR laser irradiation in vitro. **a** Owing to the decrease of NADH by ROS oxidization in the LSC + L-treated group, the cell counting kit-8 (CCK-8) assay that relies on NADH induced color change for quantification of cell survival underestimates the cell survival for the LSC + L-treated cells. **b** images of live and dead stains showing that most NCI/RES-ADR cells treated with LSC + L are alive, whereas almost all the cells are dead or detached after treated with LSC-D + L. The nanoparticle concentration was 0.02-1.6 mg ml$^{-1}$. **c–e** Viability of NCI/RES-ADR (**c**) MCF-7 (**d**), and OVCAR-8 (**e**) cancer cells after treated with free DOX, LSC nanoparticles, and DOX-laden LSC (LSC-D) nanoparticles at different concentrations (0.02–1.6 mg ml$^{-1}$ empty nanoparticle and/or 0.5–40 μg ml$^{-1}$ DOX) without or with NIR laser (L) irradiation (1 W cm$^{-2}$ for 1 min) quantified using crystal violet assay. Error bars represent s.d. ($n = 3$). **f, g** Viability of NCI/RES-ADR (**f**) and OVCAR-8 (**g**) spheres enriched with CSCs after treated with free DOX together with LSC or LSC-D nanoparticles at different concentrations (0.5–40 μg ml$^{-1}$) without or with NIR laser irradiation by using crystal violet assay. Error bars represent s.d. ($n = 3$). **h** Images of live and dead stains showing nearly all cells are alive in spheres with the LSC + L treatment. **i–j** Relative NADH in NCI/RES-ADR (**i**) and OVCAR-8 (**j**) spheres showing successful inhibition NADH in LSC-treated groups with laser irradiation (LSC + L, 1 W cm$^{-2}$, 1 min). Error bars represent s.d. ($n = 3$). The LSC + L group is compared with the LSC group with the same LSC concentration, as well as the control and laser (alone) groups. *$p < 0.05$ (Kruskal–Wallis $H$-test). The five nanoparticle concentrations (0.02, 0.04, 0.2, 0.4, and 1.6 mg ml$^{-1}$) correspond to the five concentrations of DOX (0.5, 1, 5, 10, and 40 μg ml$^{-1}$). Control cells were cultured in medium without any treatment. Scale bars: 100 μm

**Distribution and expression of P-gp efflux pumps.** We next investigated the amount and distribution of the P-gp efflux pumps in LSC + L-treated NCI/RES-ADR cells. As shown in Fig. 4a, most of the P-gp efflux pumps are located on the cell plasma membrane for control cells. Interestingly, for LSC + L-treated cells, many of the P-gp efflux pumps are distributed in the cytoplasm. This might be owing to the minimized ATP production so that there is not enough energy supply to transport the P-gp efflux pumps to the cell plasma membrane. To support this, we incubated the multidrug-resistant cells with oligomycin (an

inhibitor of ATP synthase) for 12 h to check the distribution of the P-gp efflux pumps. As shown in Fig. 4a, the P-gp efflux pumps also distributed in both the cell plasma membrane and cytoplasm, similar to the LSC + L-treated cells. These results suggest the ATP not only provides the energy for the P-gp efflux

pump to function, but also is needed to transport the P-gp efflux pumps to the cell plasma membrane. The total amount of P-gp in the multidrug-resistant cells from the aforementioned three groups was studied using flow cytometry. As shown in Fig. 4b, c and Supplementary Fig. 34, the expression of P-gp is slightly but

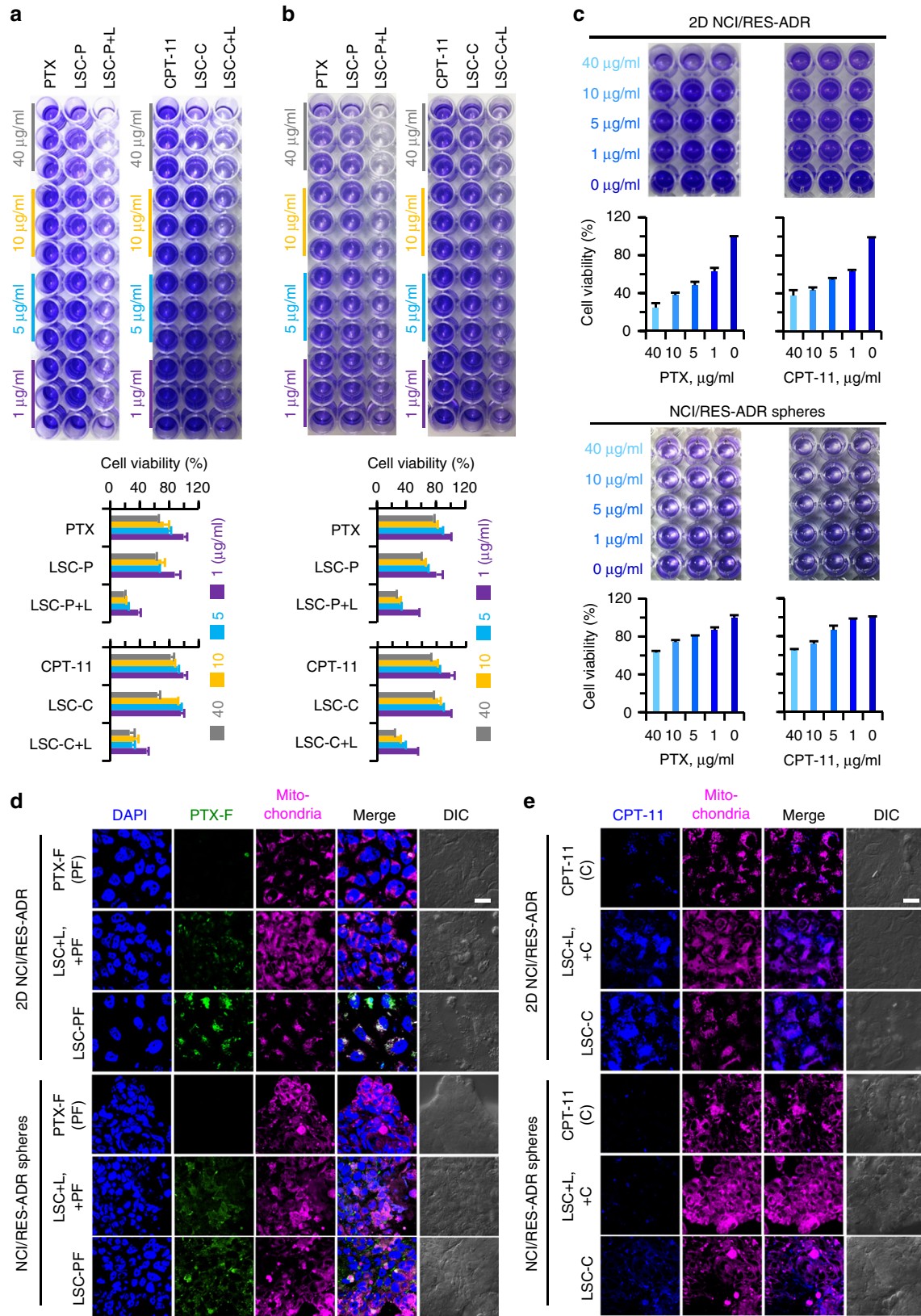

significantly decreased compared with control and oligomycin-treated cells. This together with the increased intracellular distribution of the P-gp should decrease the amount of the P-gp efflux pumps on the plasma membrane of the LSC + L-treated cells, which should also reduce the drug-resistant capacity of the multidrug-resistant cells.

**In vitro cytotoxicity**. The in vitro anticancer capability of LSC-D nanoparticles was investigated using NCI/RES-ADR, OVCAR-8, and MCF-7 cells. As shown in Supplementary Fig. 35a–c, free DOX is much less toxic to NCI/RES-ADR cells than to OVCAR-8 and MCF-7 cells, confirming the highly drug-resistant nature of the NCI/RES-ADR cells. Using LSC-D nanoparticles to deliver DOX can enhance the cytotoxicity of DOX to the multidrug-resistant cells, but it is still inefficient. When combined with NIR laser irradiation, the LSC-D nanoparticles can effectively kill all the three different types of cancer cells. However, laser irradiation alone does not compromise the viability of any of the three different cells (Supplementary Fig. 36). It is worth noting that the low cell viability associated with LSC + L treatment is misleading (Supplementary Fig. 35a–c). This is because the cell viability was quantified using the cell counting kit-8 (CCK-8) that uses NADH to generate color in the sample for quantification with colorimetry. However, the LSC + L treatment can greatly reduce NADH in cells with minimal change in the color of the CCK-8 solution (Fig. 5a). This is confirmed by the live (green) and dead (red) cell staining which shows that most of the LSC + L-treated cells are alive (Fig. 5b). In contrast, almost all of the LSC-D + L-treated cells are dead and many of them even detached from the plate at higher concentrations (Fig. 5b), consistent with the quantitative CCK-8 data. To further confirm this, crystal violet assay was used to determine cell viability. As shown in Fig. 5c–e and Supplementary Fig. 37, similar results to that shown in Supplementary Fig. 35 are obtained except for the LSC + L group.

We further checked the viability of cells in NCI/RES-ADR and OVCAR-8 spheres enriched with CSCs after various treatments using crystal violet assay. The NCI/RES-ADR spheres are not sensitive to free DOX treatment even at the highest concentration (40 μg ml⁻¹, Fig. 5f and Supplementary Fig. 38a), indicating an ultra-high drug resistance of the spheres. The OVCAR-8 spheres also show enhanced resistance to free DOX compared to their 2D-cultured counterparts (Fig. 5g and Supplementary Fig. 38a), which is in accordance with the uptake data shown in Supplementary Figs. 13 and 16. Importantly, the LSC-D + L-treated group exhibited the highest cytotoxicity. Similarly to the impact on the 2D cell cultures when using CCK-8 assay (Supplementary Fig. 35), the low cell viability of the LSC + L-treated spheres quantified using the CCK-8 assay (Supplementary Fig. 38b) is owing to the decrease of NADH and the spheres are

highly viable according to live and dead staining (Fig. 5h). Quantitative data showing the decrease of NADH but not the sum of NADH and NAD⁺ in both NCI/RES-ADR and OVCAR-8 spheres after LSC + L treatment are given in Fig. 5i, j and Supplementary Fig. 39. As discussed above for 2D cell cultures (Fig. 3c, d), this decrease in NADH is owing to the ROS produced by LSC nanoparticles in mitochondria after NIR laser irradiation (Supplementary Fig. 40).

To further confirm the capability of the LSC nanoparticles under NIR laser irradiation in overcoming multidrug resistance, two more chemotherapy drugs (paclitaxel or PTX in short and irinotecan or CPT-11 in short) were separately encapsulated in the LSC nanoparticles with an encapsulation efficiency of 50.1 ± 0.5 (for PTX) and 52.4 ± 3.7 (for CPT-11). As shown in Fig. 6a, b, the PTX or CPT-11 laden LSC nanoparticles (LSC-P or LSC-C) after irradiated with NIR laser could induce higher cytotoxicity than the free drugs or drug-laden nanoparticles alone, to both 2D-cultured NCI/RES-ADR cells and 3D-cultured NCI/RES-ADR spheres. To further confirm that this is due to the suppression of drug resistance, both 2D-cultured NCI/RES-ADR cells and 3D-cultured NCI/RES-ADR spheres were incubated with empty LSC nanoparticles. After irradiated with NIR laser, the cells were further cultured with free PTX and CPT-11 for 24 h. As shown in Fig. 6c, the toxicity of free PTX and CPT-11 to the 2D- and 3D-cultured cells exhibits a dose-dependent manner and is significantly higher than that of the two free drugs to the cells without the pre-treatment (Supplementary Fig. 41). This suggests the drug resistance capability of the cells is suppressed by the LSC + L treatment and free drugs can enter the cells. Indeed, the cellular uptake data (Fig. 6d, e) show that more free PTX (labeled with FITC as previously reported[37]) and CPT-11 (with blue fluorescence) can enter the 2D- and 3D-cultured cells (LSC + L, +PF and LSC + L, +C), although the highest drug fluorescence can be seen in the cells treated with the encapsulated drugs (i.e., LSC-PF and LSC-C). The latter indicates the advantage of using nanoparticles for drug delivery. These data further confirm the drug resistance of the NCI/RES-ADR cells can be overcome with the treatment of LSC nanoparticles and NIR laser irradiation, to enhance the antitumor capacity of the chemotherapy drugs.

**In vivo biodistribution and antitumor capability**. We next investigated the biodistribution of LSC and SC nanoparticles in mice by encapsulating indocyanine green (ICG, an NIR dye) in the nanoparticles to form LSC-I and SC-I nanoparticles, respectively. As shown in Fig. 7a, at 3 h after intravenous injection, ICG fluorescence is observable over most of the animal body for all the formulations. Importantly, enhanced ICG fluorescence in the tumor area (arrow) is observable only for mice treated with the

**Fig. 6** Enhanced anticancer capacity of LSC-P and LSC-C nanoparticles under NIR laser irradiation in vitro. **a–b** Photographs and quantitative data of crystal violet assay of 2D-cultured NCI/RES-ADR multidrug-resistant cells **a** and CSC-enriched NCI/RES-ADR spheres **b** after treated with free paclitaxel (PTX), free irinotecan (CPT-11), LSC-C nanoparticles, or LSC-P nanoparticles without or with NIR laser (L) irradiation (LSC-C + L or LSC-P + L, 1 W cm⁻² for 1 min). Error bars represent s.d. (n = 3). **c** Both 2D-cultured NCI/RES-ADR cells and CSC-enriched NCI/RES-ADR spheres were cultured with empty LSC nanoparticles (0.2 mg ml⁻¹) for 12 h. After laser irradiation (1 W cm⁻² for 1 min), the cells were further cultured with free PTX or CPT-11 dissolved (with the aid of 1% DMSO) in fresh medium for 24 h, showing the free PTX or CPT−11 could induce more cytotoxicity compared with the free drugs treated cells in **a–b**. This is probably owing to the inhibition of drug resistance ability of the cells after the combined treatment of LSC nanoparticles and laser (LSC + L). Error bars represent s.d. (n = 3). **d–e** Distribution of PTX-F (PTX labeled with fluorescein isothiocyanate (**d**) or CPT-11 (**e**) in free PTX-F or CPT−11, LSC + L treatment with free CPT-11 or PTX-F (LSC + L, + C or LSC + L, + PF), and LSC-C or LSC-PF treated multidrug-resistant cells. The results of both CPT-11 and PTX-F show that more drug could be taken up by the multidrug-resistant cancer cells after treated with LSC + L, which further confirms the drug-resistant capacity of the multidrug-resistant cancer cells can be overcome by treating the cancer cells with LSC nanoparticles and laser. Moreover, the highest drug fluorescence can be seen in multidrug-resistant cells of the LSC-C and LSC-PF groups, indicating more drugs can be delivered into the multidrug-resistant cancer cells by encapsulating the drugs in the LSC nanoparticles. The merged views of the fluorescence of CPT-11 and PTX-F with mitochondria further confirm the capability of the LSC nanoparticles in targeting mitochondria. LSC-P: paclitaxel (PTX) laden LSC nanoparticles and LSC-C: irinotecan (CPT-11) laden LSC nanoparticles. Scale bars: 20 μm

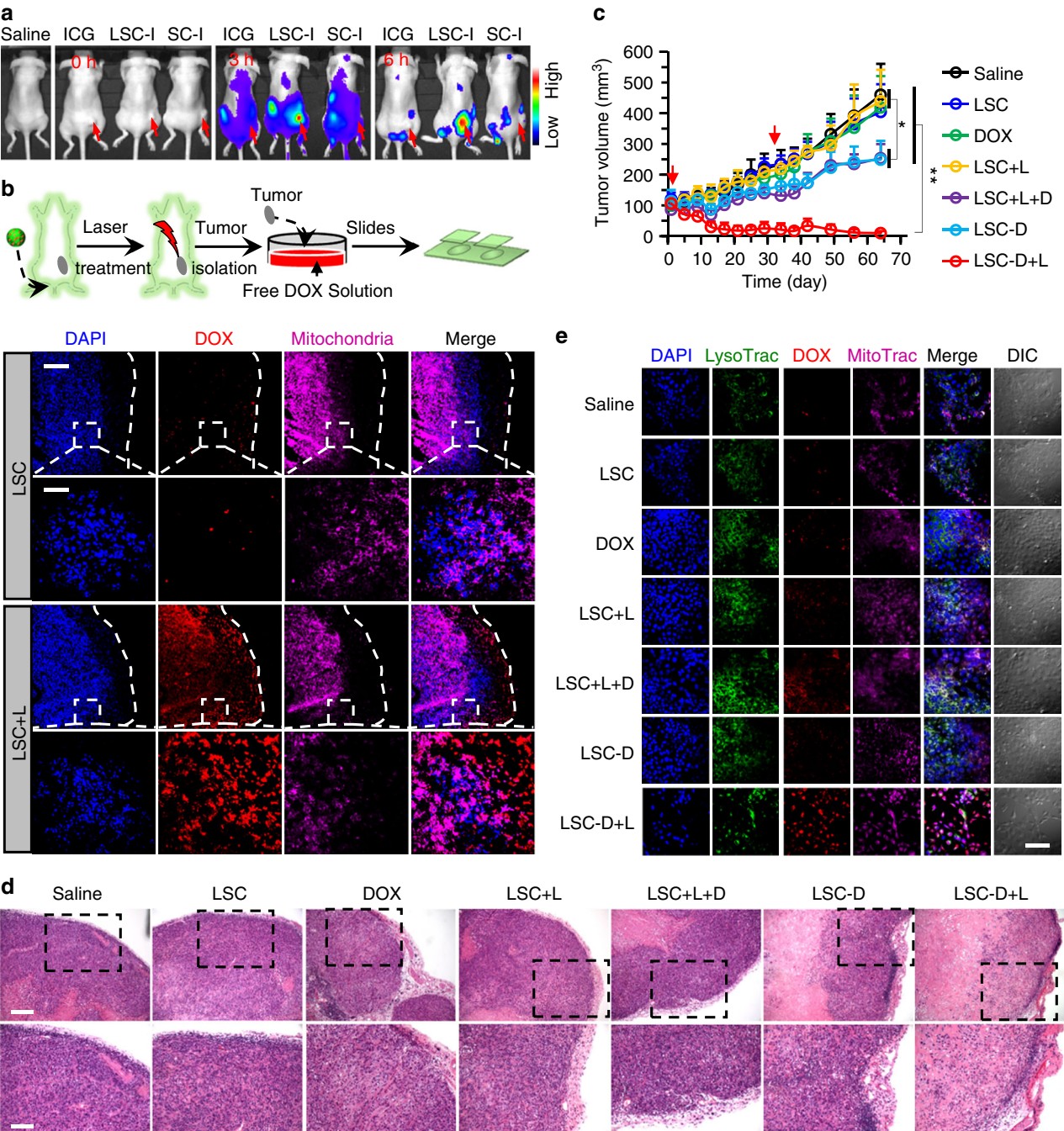

**Fig. 7** Overcoming cancer drug resistance and augmenting cancer destruction in vivo. **a** In vivo whole-animal imaging of indocyanine green (ICG) fluorescence at different times after intravenous injection via the tail vein in the form of free ICG, ICG-laden LSC (LSC-I) nanoparticles, and ICG-laden SC (SC-I) nanoparticles. The arrows indicate the locations of tumors in mice. **b** Fluorescence images of DOX in tumors of mice killed at 24 h post injection with LSC nanoparticles (with and without laser irradiation right before killing the mice). The data suggest the capability of overcoming cancer drug resistance could be achieved in vivo with the LSC + L treatment. The dashed lines indicate the boundary of the tumor tissue. The boxed regions are shown in higher magnification. Scale bar: 200 and 50 μm for low and high magnification images, respectively. **c** Growth curves of tumors in mice with various treatments showing augmented antitumor efficacy of the LSC-D + L treatments. Error bars represent s.d. ($n = 7$). $**p < 0.01$, $*p < 0.05$ (Kruskal–Wallis $H$-test). **d** Representative images of hematoxylin and eosin (H&E)-stained tumor tissue collected after killing the mice on day 64 after the first drug administration. Scale bar: 100 and 50 μm for low and high magnification images, respectively. **e** Fluorescence images of DOX in cells isolated form tumors collected on day 64 from mice with different treatments. The cells were incubated with DOX for 3 h. The data show that the drug-resistant capability of the multidrug-resistant cells in the tumors with the LSC-D + L treatment is still not fully recovered. Scale bar: 80 μm. All non-specified NIR laser (L) irradiation was at 1 W cm$^{-2}$ for 2 min. Detached NCI/RES-ADR sphere cells enriched with CSCs were used to obtain xenograft tumors for all in vivo studies

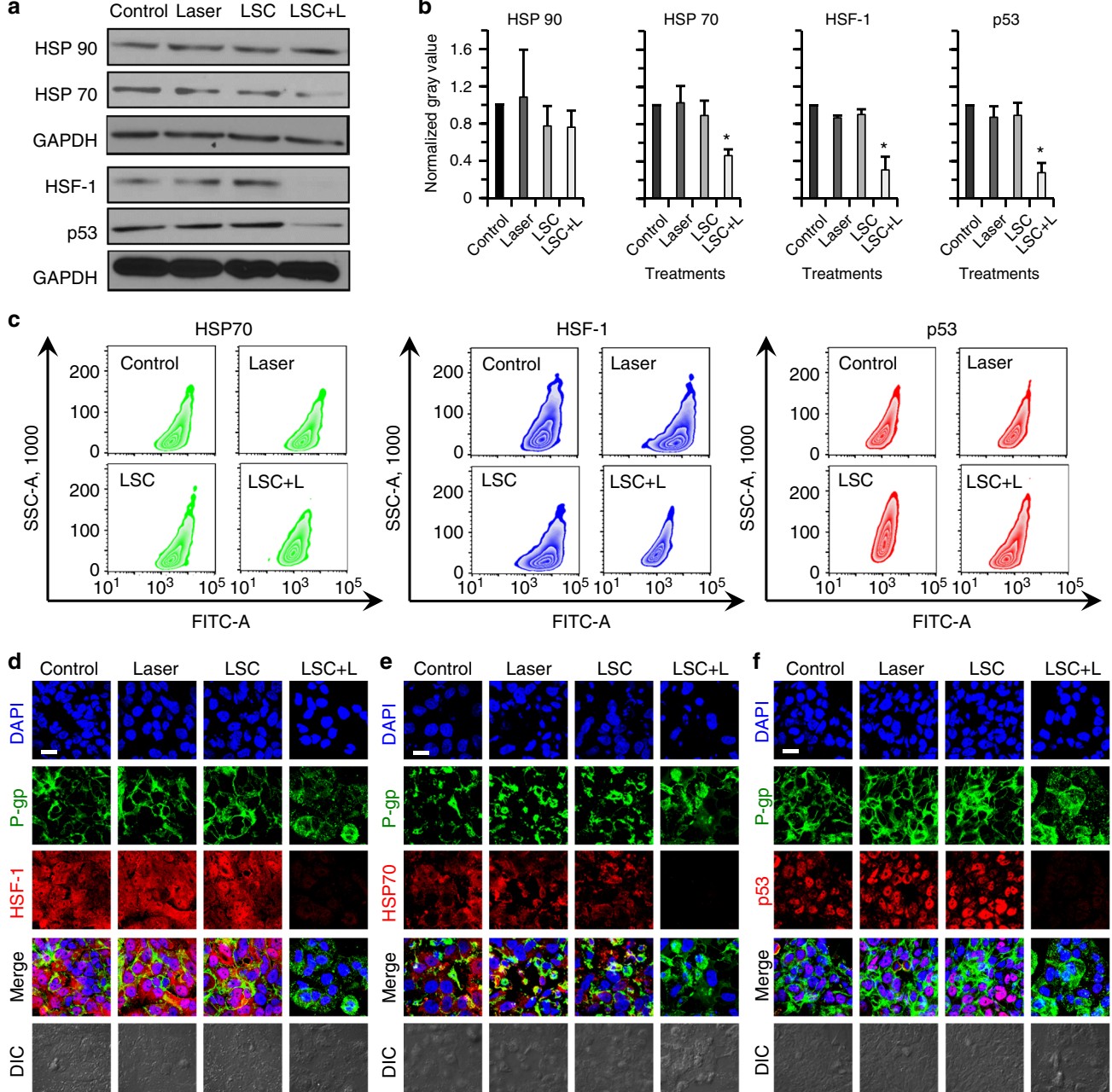

**Fig. 8** Possible mechanisms of the LSC + L treatment in sensitizing the drug-resistant cells to cancer therapy. **a** Qualitative western blot data of heat shock protein 90 and 70 (HSP90 and HSP70), heat shock factor-1 (HSF-1), and mutant p53 proteins, showing decreased expression of HSP70, HSF-1, and mutant p53 after the treatment of LSC nanoparticles and NIR laser irradiation (LSC + L). **b** Quantitative data of the western blot results, showing the decreased expression of HSP70, HSF-1, and mutant p53 in the NCI/RES-ADR cells with the LSC + L treatment is statistically significant. Error bars represent s.d. ($n = 3$). *$p < 0.05$ (Kruskal–Wallis H-test). **c** Flow cytometry analysis showing decreased expression of HSP70, HSF-1, and mutant p53 in NCI/RES-ADR cells treated with LSC nanoparticles and NIR laser irradiation. **d** Confocal images of HSF-1 showing the HSF-1 distributes in both nuclei and cytoplasm. **e** Confocal images of HSF70 showing the HSF70 mainly distributes in cytoplasm. **f** Confocal images of mutant p53 show the mutant p53 mainly distributes in nuclei. The distribution of P-gp is similar to that shown in Fig. 4a. No treatment (laser or nanoparticle) was conducted on the cells in the control group. The cells were permeabilized for the flow cytometry and confocal analyses. The NIR laser irradiation was at 1 W cm$^{-2}$ for 1 min. The concentration of LSC nanoparticles used for the LSC or LSC + L groups was 0.2 mg ml$^{-1}$. Scale bars: 20 μm

LSC-I nanoparticles, suggesting the tumor targeting capacity of LSC nanoparticles. In mice treated with LSC-I nanoparticles, the ICG fluorescence in the tumor area further increases at 6 h. In contrast, no ICG fluorescence is observable at 6 h in the tumor area of mice treated with either free ICG or SC-I nanoparticles. To confirm the whole-animal imaging data, various organs were harvested for ex vivo imaging after sacrificing the mice at 6 h.

Indeed, strong fluorescence of ICG can only be detected in the tumor from LSC-I-treated mouse (Supplementary Fig. 42).

To examine the capability of overcoming cancer drug resistance of LSC nanoparticles in vivo, mice either with or without NIR irradiation treatment in the tumor area were scarified at 24 h after intravenous injection of LSC nanoparticles. The tumors collected from the killed mice were incubated in

medium with DOX for 3 h and then cryo-sectioned for examination with fluorescence microscopy. As shown in Fig. 7b, the tumors are still highly drug-resistant because DOX is not clearly observable in the tumors of mice treated with LSC nanoparticles alone. More importantly, DOX does enter the cells in the tumors from mice treated with LSC + L. These observations indicate that the combination of LSC nanoparticles and NIR irradiation can overcome cancer drug resistance in vivo.

At last, the safety and antitumor efficacy of the LSC-D nanoparticles were investigated in vivo. Tumor growth for treatments with LSC nanoparticles, free DOX, or LSC + L is similar to that of saline control (Fig. 7c). A decrease in tumor volume is observable for treatments of LSC-D nanoparticles, and LSC + L in combination with free DOX (LSC + L + D). The latter is probably owing to the capability of overcoming cancer drug resistance by the treatment of LSC + L although free DOX does not preferentially accumulate in the tumor. Importantly, for LSC-D + L treatment, complete tumor destruction was observed in two of seven mice, and both size and weight of the tumors in the LSC-D + L group were significantly less than that of other treatment groups (Fig. 7c and Supplementary Fig. 43). Moreover, hematoxylin and eosin (H&E) stain shows extensive necrosis in the tumors from the LSC-D + L group, whereas tumors from all the other groups are more viable (Fig. 7d). We further isolated cells from tumors of all the treatment groups. After growing for 3 days, the cells were incubated with free DOX in medium for 3 h. As shown in Fig. 7e, cells from tumors treated with saline, LSC, DOX, LSC-D, and LSC + L are drug resistant. Compared with tumors treated with LSC + L, more DOX can be seen in cells isolated from the LSC + L + D-treated tumors, suggesting the toxicity of DOX to tumor cells can delay the recovery of their drug resistance. Indeed, DOX fluorescence is strongest in cells isolated from tumors treated with LSC-D + L because more DOX can be delivered into tumor with the LSC-D than LSC + L + D treatment (Fig. 7a).

The body weights of mice were stable during the 64 days of observation (Supplementary Fig. 44a), probably because we treated the mice only twice at a low dose of DOX (2.5 mg kg$^{-1}$ body weight) and more than 1-month apart. In addition, there is no obvious damage in the H&E stain of major organs collected from LSC-D + L-treated mice (Supplementary Fig. 44b). Taken together, these data suggest the LSC-D nanoparticles are safe for in vivo treatment.

## Discussion

To understand the possible molecular mechanisms of the LSC + L treatment in sensitizing the drug-resistant cells to cancer therapy, we investigated the expression of two stress-related proteins in the NCI/RES-ADR cells first. As shown in Fig. 8a, b, the expression of the 70-kDa heat shock protein (HSP70) is significantly decreased at 12 h after the LSC + L treatment compared with control, NIR laser, or LSC nanoparticles treated cells although no significant change is observable for the expression of 90-kDa heat shock protein (HSP90). HSP70 is abnormally overexpressed in majority of cancer cells and important for cancer cell survival[38]. Therefore, the decrease in HSP70 could make the cancer cells more sensitive to chemotherapy drugs[39]. Moreover, the expression of heat shock factor-1 (HSF-1) that is commonly thought to be associated with HSP70[40], is also decreased at 12 h after the LSC + L treatment. Previous studies have shown that high HSF-1 expression is associated with poor outcome of chemotherapy (including lung, breast, and colon cancer) and increases CSC frequency[41,42]. Therefore, the decreased expression of HSF-1 is beneficial for chemotherapy of not only regular cancer cells but also the CSCs. It is also worth noting that the

HSP70 and HSF-1 protein complex is required for the stabilization of mutant p53 protein in cancer cells[43]. The decrease of both HSP70 and HSF-1 should compromise the stability of the mutant p53 protein. Indeed, the mutant p53 protein is highly expressed in the NCI/RES-ADR cells and the LSC nanoparticles or NIR laser irradiation alone has no impact on its expression. Importantly, it is significantly decreased in the LSC + L-treated cells. Similarly, the mutant p53 is important to the resistance of cancer cells to both apoptosis and chemotherapy[44]. We further confirmed the aforementioned observations from western blotting studies using flow cytometry (Fig. 8c). Confocal fluorescence images show that the HSF-1 is distributed in both the cytoplasm and nuclei of cells in the control, NIR laser, or LSC groups, but decreased in the LSC + L group (Fig. 8d). HSP70 mainly distributes in the cytoplasm, whereas mutant p53 proteins are mainly observed in nuclei of cells for the control, laser, or LSC groups, and both are decreased in LSC + L-treated cells (Fig. 8e, f). Previous studies show that the expression of P-gp and mutant p53 shares the same trend in clinical samples[45]. In addition, both HSP70 and HSF-1 have been shown to regulate the expression of P-gp in cancer cells[46,47]. Therefore, in this study, the decreased expression of P-gp is possibly owing to the decreased expression of HSP70, HSF-1, and mutant p53.

In summary, we carefully designed a novel lipid, carbon, and silica hybrid nanoparticle (Supplementary Note 10), which can be used to target mitochondria and produce ROS in mitochondria under NIR irradiation to oxidize NADH for inhibiting the production of ATP. This compromises the function of efflux pumps and overcomes the drug resistance of multidrug-resistant NCI/RES-ADR cancer cells and their CSCs both in vitro and in vivo. Ultimately, the drug-laden LSC nanoparticles exhibit superior safety and efficacy for therapy of multidrug-resistant tumors. This study provides a promising nanotechnology-based strategy for fighting against cancer multidrug resistance.

## Methods

**Materials**. Cyclohexane, hexanol, triton X-100, toluene, TEOS, APTMS, and ICG were purchased from Sigma (St. Louis, MO, USA). DPPC and cholesterol were purchased from Anatrace (Maumee, OH, USA). DOX was purchased from LC laboratories (Woburn, MA, USA). Irinotecan (CPT-11) was purchased from Selleck Chemicals (Houston, TX, USA). The CCK-8 cell proliferation reagent was purchased from Dojindo Molecular Technologies (Rockville, MD, USA). Fetal bovine serum (FBS) and penicillin/streptomycin were purchased from Invitrogen (Carlsbad, CA, USA). The RPMI 1640 and EMEM cell culture media were purchased from ATCC (Manassas, VA, USA). All other chemicals were purchased from Sigma unless specifically mentioned otherwise.

**Synthesis of nanoparticles**. CCS nanoparticles were prepared as published before[48]. In brief, 6 g of glucose was dissolved in 200 ml of deionized water, followed by stirring and sonication to obtain a homogeneous solution. The colorless solution was transferred into a Teflon stainless steel autoclave with 500 ml capacity and sealed closely. Subsequently, the sealed autoclave was heated to 180 °C for 10 h with constant stirring at ~800 rpm, and then cooled to room temperature passively. Finally, the suspension containing the as-prepared CCS was transferred into a flask for further characterization and use. To prepare SC nanoparticles, distilled water suspended with CCS nanoparticles was mixed with hexanol (5 ml), triton X-100 (1.7 ml), and cyclohexane (2 ml). After added 60 μl of ammonium hydroxide (28 wt%) and 100 μl of TEOS, the mixture was stirred at 800 rpm using a mini-stir bar for 3, 6, or 12 h at room temperature. At last, the reaction was stopped by adding 30 ml of ethanol. The nanoparticles were collected by centrifuging at 13,800 g for 10 min and washed with ethanol and water for two times. The APTMS-coated SC nanoparticles were obtained by incubating the nanoparticles and 20 μl of APTMS in 3 ml of ethanol and stirring at 200 rpm using a mini-stir bar at room temperature for 12 h. The preparation and collection of LSC nanoparticles were conducted as previously reported[25]. LS nanoparticles were prepared with the same method as that for preparing LSC nanoparticles except that distilled water was used instead of the CCS nanoparticle suspension during the preparation of SC nanoparticles.

**Characterization of nanoparticles**. The size distribution of nanoparticles (1 mg ml$^{-1}$ in deionized water) was measured using a Brookhaven 90 Plus/BI-MAS DLS instrument. FTIR spectroscopy analysis of the nanoparticles was conducted using a

Perkin Elmer (Waltham, MA, USA) Spectrum 100 FTIR spectrometer according to the manufacturer's instruction. In brief, dried samples were grounded in an agate mortar, mixed with KBr at a ratio of ~1:80 (nanoparticle:KBr) in weight, and pressed into small discs at 10 tons for 5 min. For proton nuclear magnetic resonance ($^1$H NMR), 2 mg of CCS, SiO$_2$, SC, or LSC nanoparticles were suspended in 1 ml of heavy water (D$_2$O) and characterized using a Bruker 400 MHz spectrometer (Billerica, MA, USA). For TEM characterization, the nanoparticles were imaged directly or stained with uranyl acetate solution (2%, w/w, for LSC nanoparticles) using an FEI (Moorestown, NJ, USA) Tecnai G2 Spirit transmission electron microscope, as detailed elsewhere[25]. For SEM characterization, 10 µl of nanoparticles in aqueous solutions were dropped on a freshly cleaved mica grid and dried for 60 min in air. A thin film of Au was then sputter-coated onto the nanoparticles on the substrate. Samples were imaged with an FEI NOVA Nano400 scanning electron microscope.

**Detection of singlet oxygen.** Solutions of LSC nanoparticles with or without laser irradiation were mixed with Singlet Oxygen Sensor Green reagent (Life Technologies, Carlsbad, CA, USA) and fluorescence (excitation and emission at 488 and 525 nm, respectively) were measured using a Jasco (Easton, MD, USA) FP-6200 spectrofluorometer.

**Detection of pyruvate in nanoparticles.** The amount of pyruvate in nanoparticles was measured by the Pyruvate Assay Kit (Sigma). In brief, 50 µg of CCS, SiO$_2$ nanoparticles, SC nanoparticles (reaction times of 3 h and 6 h), or LSC nanoparticles were dissolved in 50 µl of DI water and mixed with Master Reaction Mix (50 µl containing: Pyruvate Assay Buffer, 46 µl; Pyruvate Probe Solution (colorimetric), 2 µl; and Pyruvate Enzyme Mix, 2 µl) at room temperature for 30 min in dark. The amount of pyruvate was then quantified according to the manufacturer's instruction using a Perkin Elmer VICTOR X4 multilabel plate reader.

**TEM imaging of cells.** Cells were seeded into Nunc Lab-Tek II Chamber Slide System (Thermo Fisher Scientific Inc., Waltham, MA, USA) at a density of $1 \times 10^6$ cells ml$^{-1}$ and incubated in medium without (control) or with LSC-D nanoparticles for 9 h with or without laser irradiation (1 W cm$^{-2}$ for 1 min). Samples were prepared for TEM according to standard procedures and imaged using an FEI (Moorestown, NJ, USA) Tecnai G2 Spirit transmission electron microscope.

**Detection of reactive oxygen species in cells.** NCI/RES-ADR cells were incubated with nanoparticles for 24 h with or without NIR laser irradiation. The cells were changed with fresh medium containing 25 µM dichloro-dihydro-fluorescein diacetate (DCFH-DA, Cell Bio Labs Inc., San Diego, CA, USA) and incubated for 45 min. The production of reactive oxygen species (ROS) was measured by fluorescence intensity using a PerkinElmer (Waltham, MA, USA) VICTORX3 Multilabel microplate reader and studied qualitatively by fluorescence microscopy.

**Detection of NADH and NAD$^+$ in cells.** NCI/RES-ADR cells (or spheres, ~ $2 \times 10^5$) were incubated with LSC nanoparticles for 12 h. After irradiated with NIR laser for 1 min, cells were harvested and washed with cold PBS. Cell pellets were obtained by spinning at 2300 g for 5 min and removing the supernatant. Then, 400 µl of NADH/NAD$^+$ Extraction Buffer (NAD$^+$/NADH Assay Kit (Colorimetric), Abcam, Cambridge, MA, USA) were added to re-suspend the cells and the resultant samples were frozen/thawed (20 min on dry ice followed by 10 min at RT) twice. After mixing for 10 s by vortex, the samples were centrifuged at 4 °C for 5 min. Supernatant (containing extracted NAD$^+$/NADH) was then collected and transferred into a new tube. For NADH detection, the supernatant was heated at 60 °C for 30 min in water bath to decompose NAD$^+$. A total of 50 µl of the heated supernatant was mixed with NAD Cycling Buffer (98 µl) and NAD Cycling Enzyme Mix (2 µl) for 5 min at room temperature. For detection of the sum of NAD$^+$ and NADH, the supernatant was used directly. A total of 10 µl of NADH Developer was added and the sample was kept at room temperature for 4 h. The mixture was then detected quantitatively using a PerkinElmer (Waltham, MA, USA) VICTORX3 Multilabel microplate reader at 450 nm.

**Detection of ATP consumption in cell plasma membrane.** Cells were collected by centrifugation at 600 g for 5 min at 4 °C. After washed with 1 ml of ice-cold PBS, cells were re-suspended in 1 ml of the Homogenize Buffer Mix (Plasma Membrane Protein Extraction Kit, Abcam, Cambridge, MA, USA) in an ice-cold Dounce homogenizer (Corning, Lowell, MA, USA). After transferred the homogenate into a 1.5 ml microcentrifuge tube, the mixture was centrifuged at 700 g for 10 min at 4 °C and the supernatant was further centrifuged at 10,000 g for 30 min at 4 °C. The resultant pellet contains the total membrane proteins. Then, the pellet was mixed with 0.1 mM ATP for 1 h and centrifuged at 10,000 g for 30 min at 4 °C to collect the supernatant. The amount of ATP in the supernatant was measured by an ATP Assay Kit (Abcam, Cambridge, MA, USA). Briefly, 50 µl of supernatant was mixed with ATP Reaction Mix (ATP Assay Buffer: 44 µl; ATP Probe: 2 µl; ATP Converter: 2 µl; and Developer Mix: 2 µl) and incubated at room temperature for 30 min in dark. The amount of ATP in the mixture was then detected quantitatively using a PerkinElmer (Waltham, MA, USA) VICTORX3 Multilabel microplate reader at 570 nm.

**Encapsulation of theranostic agents and drug release.** Encapsulation of DOX or ICG was conducted by mixing it with the nanoparticles at a feeding ratio of 1:20 (DOX or ICG: nanoparticles) in deionized (DI) water for 24 h. To encapsulate CPT-11, paclitaxel (PTX), or fluorescein isothiocyanate (FITC)-labeled PTX (PTX-FITC), the (3-Aminopropyl) trimethoxysilane (APTMS)-SC nanoparticles were mixed with the agent at 1:20 (CPT-11, PTX, or PTX-FITC: nanoparticles) in ethanol for 24 h. After removing ethanol by evaporation, the nanoparticles were coated with lipid membrane. The encapsulation efficiency (EE) was calculated using the following equation:

$$\mathrm{EE} = W_{\mathrm{encapsulated}} / W_{\mathrm{fed}} \times 100\% \qquad (1)$$

where $W_{\mathrm{encapsulated}}$ represents the amount (in weight) of DOX, CPT-11, PTX, or ICG encapsulated into nanoparticles and $W_{\mathrm{fed}}$ is the initial total amount of DOX, CPT-11, PTX, or ICG fed for encapsulation. The amount of DOX, CPT-11, PTX/PTX-FITC, and ICG was determined spectrophotometrically using a Beckman Coulter (Indianapolis, IN, USA) DU 800 UV-Vis spectrophotometer based on their absorbance at 483, 370, 280, and 780 nm, respectively.

To investigate the drug release, drug-laden nanoparticles (20–30 mg) were dissolved in PBS (5 ml, at pH 7.4) and transferred into a dialysis bag (MWCO: 20 kDa) that was placed in 30 ml of the same PBS solution and stirred at 110 rpm using a mini-stir bar at 37 °C. For measurement, dialysate was collected and replenished with same volume of fresh PBS. The released DOX from nanoparticles was measured using UV-Vis spectrophotometry. For NIR laser irradiation-triggered drug release, the nanoparticle solution was centrifuged at 13,800 g to obtain the supernatant. The DOX concentration in the supernatant was analyzed using UV-Vis spectrophotometry.

**Electron paramagnetic resonance spectroscopy.** Electron paramagnetic resonance (EPR) measurements were performed on a continuous wave X-band Bruker EMXPlus EPR spectrometer. Each sample was dissolved in DI water and immediately transferred into a glass capillary tube that was placed in the microwave cavity of the EPR spectrometer. The applied NIR laser irradiation was at 1 W cm$^{-2}$ for 1 min, where specified. The concentrations of DEPMPO and LSC nanoparticles were 20 mM and 2 mg ml$^{-1}$, respectively. All EPR spectra were recorded at room temperature under identical conditions, using 20 mW microwave power, 0.1 G field modulation at 100 kHz, and 16 scans.

**Measurement of hydroxyl radical with terephthalic acid.** Terephthalic acid (TA) or 2-hydroxy terephthalic acid (2-HTA) was dissolved in DI water without or with LSC nanoparticles (2 mg ml$^{-1}$). The concentration of TA or 2-HTA was kept at 82.1 nM for all the measurements. After irradiated without or with laser for 1 min (1 W cm$^{-2}$), the mixture was measured (excitation: 315 nm) using a JASCO FP-6200 spectrofluorometer (MD, USA).

**Cell culture and in vitro cell viability.** Human multidrug-resistant NCI/RES-ADR cancer cells (human ovarian cancer cell line that was called MCF-7/ADR in early studies, from ATCC and used without further authentication or test for mycoplasma contamination) and human breast cancer MCF-7 cells (ATCC) were cultured in EMEM supplemented with 10% FBS and 1% penicillin/streptomycin at 37 °C in a humidified 5% CO$_2$ incubator. Human ovarian cancer OVCAR-8 cells (ATCC) were cultured similarly in RPMI 1640 medium supplemented with 10% FBS and 1% penicillin/streptomycin. The NCI/RES-ADR cells were used as the drug-resistant cell model with the MCF-7 and OVCAR-8 cells as the control non-drug-resistant cell models. For in vitro cell viability studies, OVCAR-8, MCF-7, and NCI/RES-ADR cells were seeded and incubated with various drug formulations for 12 h. Free DOX and nanoparticles could dissolve in medium directly while free CPT-11 and PTX were dissolved in dimethyl sulfoxide (DMSO) and then mixed with medium at the ratio of 1: 100 (DMSO: medium). Then, the medium was replaced with PBS and if needed, the cells were further treated with near infrared (NIR) laser irradiation (at 1 W cm$^{-2}$ for 1 min). After treated with laser, cells were further cultured with various drug formulations for 12 h. The cell viability was then evaluated using the CCK-8 assay according to the instruction given by the manufacturer (Dojindo Molecular Technologies, Rockville, MD, USA). Cell viability was calculated as the ratio of the cell number determined for each group to that of control group with no treatment.

To obtain cancer cell spheres enriched with CSCs, the well-established suspension culture was used[33–35]. Briefly, detached single cancer cells were collected and cultured with CSC medium in 24-well ultralow attachment plates (Corning, Lowell, MA, USA) at a density of 20,000 cells ml$^{-1}$. The CSC medium consisted of DMEM/F12-K supplemented with 5 µg ml$^{-1}$ insulin, 20 ng ml$^{-1}$ epidermal growth factor (EGF), 20 ng ml$^{-1}$ basic fibroblast growth factor (bFGF), $1 \times$ B27 (Invitrogen), and 0.4% (w v$^{-1}$) bovine serum albumin (BSA). The spheres were cultured in CSC medium for 10 days and then collected for further experimental use. To determine cell viability in the spheres, the OVCAR-8 and NCI/RES-ADR spheres obtained after 10-day suspension culture in the CSC medium in 24-well plates were further incubated for 12 h in the CSC medium containing various drug formulations either with or without near infrared (NIR) laser irradiation (at 1 W cm$^{-2}$ for 1 min). The cell viability was then evaluated with the CCK-8 assay in the same way as that for the 2D-cultured cells.

For crystal violet assay, cells were treated in the same way as aforementioned, and then fixed with 10% methanol and stained with 0.1% crystal violet (dissolved in 10% methanol). After staining, cells were washed three times with PBS and de-stained with acetic acid. The absorbance of the crystal violet solution was measured at 590 nm and compared with a standard curve of the known cell numbers versus the absorbance for interpolation, to determine the cell number in the samples. The cell numbers under various experimental conditions were normalized to that of the control condition (in medium all the time) to obtain cell viability.

**Flow cytometry analysis**. The NCI/RES-ADR cells were incubated in medium containing various drug formulations for 12 h. After irradiated with laser for 1 min at 1 W cm$^{-2}$ (if needed), cells were further cultured for 12 h, washed twice with 1 × PBS at 4 °C, detached using trypsin/ethylenediaminetetraacetic acid (EDTA), and fixed with 4% paraformaldehyde for 20 min at room temperature. After washing, the fixed cells were incubated in 3% BSA and 0.1% TritonX-100 in 1 × PBS at room temperature for 1 h to block potential nonspecific binding and permeabilize the cell plasma membrane, respectively. Following that, the fixed and permeabilized cells were incubated overnight at 4 °C with HSP90 antibody (Cell Signaling, 4874S), HSP70 antibody (Cell Signaling, 4872S), mutant p53 antibody (Abcam, ab32049), HSF-1 antibody (ThermoFisher, PA3–017), and P-gp antibody (Sigma, P7965) at the dilution ratio of 1:200. Unbounded antibody was washed away with 1 × PBS for 3 times. Cells were then incubated with secondary antibody (ThermoFisher) at the dilution ratio of 1:200 in 1 × PBS with 1% BSA at room temperature for 1 h. After washing with 1 × PBS twice, the cells were analyzed using a BD (Franklin Lakes, NJ, USA) LSR-II flow cytometer and Diva software.

**In vitro imaging**. Collagen-coated cover glasses (Nunc, Thermo Fisher Scientific Inc.) were placed at the bottom in six-well plate. Cancer cells were then collected and cultured on the cover glasses in the plate at a density of 2 × 10$^5$ cells per well at 37 °C for 12 h. The medium was then replaced with 2 ml of fresh medium containing different drug formulations. After incubation at 37 °C for the desired time, cells were treated with medium containing LysoTracker Green DND-99 (Life Technologies) and/or MitoTracker Deep Red (Life Technologies) to stain late endosomes/lysosomes and/or mitochondria, respectively. DOX and CPT-11 have intrinsic fluorescence and can be imaged directly. For imaging of PTX, it was labeled with FITC using a previously reported method[37]. In brief, FITC (5 mg) and PTX (50 mg) were dissolved in DMSO (2 ml) at 90 °C for 2 h in darkness. After removing DMSO at 90 °C by rotary evaporation for ~12 h, the product was washed with DI water by centrifugation/precipitation (PTX with and without labeling did not dissolve in water, whereas FITC did) for three times to remove unreacted FITC. For P-gp, HSP70, HSF-1, and mutant p53 imaging, cells cultured on collagen-coated cover glasses were fixed with 4% paraformaldehyde for 20 min at room temperature. After washing, fixed cells were incubated in 3% BSA and 0.1% TritonX-100 in 1 × PBS at room temperature for 1 h to block potential nonspecific binding and permeabilize the cell plasma membrane, respectively. Following that, the fixed and permeabilized cells were incubated overnight at 4 °C with the same antibodies used in the aforementioned flow cytometry studies at the dilution ratio of 1:200. Unbounded antibody was removed by washing with 1 × PBS for three times. Cells were then incubated with secondary antibody at the dilution ratio of 1:200 in PBS with 1% BSA at room temperature for 1 h, and washed three times with 1 × PBS. The cover glass attached with cells was mounted onto a glass slide with anti-fade mounting medium (Vector Laboratories Burlingame, CA, USA) for examination using an Olympus FluoView FV1000 confocal microscope.

**Western blotting**. Cells were cultured and treated similarly to that for the aforementioned flow cytometry studies. The cells were lysed with RIPA buffer supplemented with phosphatase inhibitors and protease inhibitors. The proteins were collected and their concentrations determined with a NanoDrop 2000 Spectrophotometer (Thermo, IL, USA). The protein samples (20 μg) were separated on a 10% sodium dodecyl sulphate–polyacrylamide gel and electrophoretically transferred to polyvinylidene difluoride membranes in Tris–glycine transfer buffer. The membranes were blocked in 5% (w v$^{-1}$) instant nonfat dried milk for 2 h at room temperature, and incubated with the same primary antibodies used in the aforementioned flow cytometry studies at the dilution ratio of 1:1000 at 4 °C overnight. The membranes were subsequently washed with Tris Buffered Saline with Tween 20 (50 mM Tris–HCl, pH 7.5, 150 mM NaCl, and 0.05% Tween 20) and incubated with secondary horseradish peroxidase-conjugated IgG (1:1000) for 1 h at room temperature. Immunoreactive bands were visualized using chemiluminescence (ECL Kit; Pierce Biotechnology) and captured by a Bio-Rad molecular imager. The total intensity of the band for each protein was calculated with ImageJ and normalized to that of GAPDH. Uncropped images of the western blot data are shown in Supplementary Fig. 45.

**Animals and xenograft tumors**. Athymic male NU/NU nude mice (6-week old) were purchased from Charles River (Wilmington, MA, USA) and maintained on a 16:8 h of light–dark cycle. All procedures for animal use were approved by the Institutional Animal Care and Use Committee (IACUC) at The Ohio State University. To obtain NCI/RES-ADR xenograft in the nude mice, detached NCI/RES-ADR sphere cells were suspended in a mixture (1:1) of 1 × PBS and matrigel at 2 ×

10$^5$ cells ml$^{-1}$. A total of 20,000 cells (in 100 μl of the mixture) were subcutaneously injected at the dorsal side of the upper hind limb of each 7-week-old mouse.

**In vivo imaging and biodistribution**. When the tumor reached ~5 mm in long diameter, the mice were intravenously injected with 100 μl of saline, 50 μg of free ICG dissolved in 100 μl of saline, 50 μg of ICG encapsulated in SC nanoparticles dissolved in 100 μl of saline, or 50 μg of ICG encapsulated in LSC nanoparticles dissolved in 100 μl of saline via the tail vein. All the mice were imaged at 0 (right before injection), 3, and 6 h after injection, using a PerkinElmer (Waltham, MA, USA) IVIS instrument with excitation at 780 nm and an 831 nm filter to collect the fluorescence emission of ICG. For ex vivo imaging, the mice were killed and the major organs (liver, kidney, lung, spleen, and heart) and tumor were collected for further ex vivo fluorescence imaging of ICG using the same condition for whole body imaging.

To study the capability of overcoming cancer drug resistance in vivo, mice were injected with LSC nanoparticles and treated with or without laser irradiation at 12 h after the injection. Then, tumors were collected and cultured in medium with DOX (10 μg ml$^{-1}$) for 3 h. After washed with PBS for three times, the tumors were put in frozen with the Tissue-Tek (Sakura Finetek, Torrance, CA, USA) O.C.T. Compound and Cryomold at −80 °C for 24 h. The tumors were then cut into slices of 10 μm thick using a cryo-microtome and transferred onto microscope slides. The slides were fixed in cold acetone for 10 min and soaked in Tris buffer for 10 min. The slides were then incubated with 4',6-diamidino-2-phenylindole (1 μg ml$^{-1}$) and MitoTracker Deep Red (500 nM) in dark for 10 min to stain the cell nuclei and mitochondria, respectively. Afterward, the slides were examined using an Olympus FluoView FV1000 confocal microscope.

**In vivo antitumor efficacy and safety**. After tumors reached ~5 mm in long diameter, mice were randomly allocated to each group and treated with 100 μl of saline, free DOX, LSC nanoparticles without (LSC) or with (LSC + L) NIR laser irradiation, LSC nanoparticles with NIR laser irradiation and free DOX (LSC + L + D), LSC-D nanoparticles without (LSC-D) or with (LSC-D + L) NIR laser irradiation (n = 7). The number of mice was chosen to guarantee adequate power for analyzing the difference between the saline and LSC-D + L groups. The blinding method was not applied for this in vivo animal study. All the drug formulations were dissolved in 100 μl of saline. The dose of DOX for all the formulations with the drug was 2.5 mg kg$^{-1}$ body weight. The NIR laser irradiation was at 1 W cm$^{-2}$ for 2 min and conducted at 12 h after intravenous drug injection. In order to keep the temperature lower than ~43 °C during laser irradiation, the tumor was passively cooled at room temperature for 1 min after 1 min of laser irradiation. These treatments were repeated once on day 34 after the initial treatments. The tumor volume (V) was calculated as: $V = L × W^2 × 0.5$, where L is long diameter and W is short diameter determined using a caliper. The mice were killed on day 64 after the first treatments. Tumors, livers, lungs, hearts, spleens, and kidneys from different groups were collected and fixed in formalin for H&E staining.

**Statistical analysis**. All data are reported as mean ± standard deviation (s.d.) from at least three independent runs. The Kruskal–Wallis H-test and the Mann–Whitney U-test were used to assess the overall among-group and two-group differences (that follow normal distribution and equal variances), respectively. In all cases, a p value less than 0.05 was considered to be statistically significant.

**Data availability**. All data supporting the findings of this study are available from the corresponding authors upon request.

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

## Acknowledgements

This work was partially supported by grants from American Cancer Society (ACS #120936-RSG-11-109-01-CDD) and NIH (R01CA206366) to X.H., a Pelotonia Post-doctoral Fellowship to H.W., and grants from NSF (CMMI-1418696 and CMMI-1358673) to X.L. G.J. was supported by the National Natural Science Foundation of China (No. 81620108030). C.P.J. was funded by grant from NSF (MCB-1715174).

## Author contributions

H.W., X.H., and X.L. conceived the project. H.W. and X.H. analyzed the data and wrote the manuscript. H.W. conducted all the experiments with help from Z.G., X.Liu, P.A., S.Z., and D.W.C. X.L., X.Lu, Z.L., C.P.J., G.J., and J.Y. edited the manuscript. All authors approved the manuscript.

## Additional information

**Competing interests:** The authors declare no competing financial interests.

