## [Peer Review File · Nature Communications]

Reviewers' comments:

Reviewer #1 (Remarks to the Author):

Review

This article reports on lipid membrane-coated silica-carbon (LSC) hybrid nanoparticles that can specifically produce reactive oxygen species (ROS) in mitochondria under near-infrared (NIR) laser irradiation. The intention of these was to treat MDR cancer. The challenge for this manuscript is that the activity of the nanoparticles are empiric - the mitochondrial targeting was not a matter of design, but once observed its mechanism not characterized. The literature is awash with nanoparticle formulations of doxorubicin that are active against NCI-ADR/RES cells and other MDR cell lines, and when the focus shifts to ROS generation, the underlying cause of ROS generation is not determined, it is only characterized. The observational nature of this study leads this reviewer to conclude it is not appropriate for publication in Nature Communications, and is probably more appropriate for a cancer journal.

As an example of utility, the authors encapsulated doxorubicin into the LSC nanoparticles.

The manuscript then moves on to mitochondrial targeting. The authors argue that the nanoparticles can target mitochondria 'die to the pyruvate', and the localization is confirmed using TEM. While the localization is demonstrated, the mechanism is speculated, but not characterized. Given the negative potential at the mitochondrial membrane, a net positive charge on a small molecule is usually associated with mitochondrial accumulation - the mechanism for mitochondrial targeting would need to be elucidated to sufficiently strengthen this manuscript.

It is impossible to conclude from Fig 3B that the nanoparticles are targeting the mitochondria - there appear to be significant numbers of nanoparticles outside the mitochondria all - can the particles be counted and quantified across multiple fields of view? There appears to be an absence of free doxorubicin controls in Figure 3.

Again, it is observed that (presumably) P-gp is not active in cells treated with nanoparticles (yet another mechanism of action for MDR sensitization?) - is this due to the ROS generation? Or a specific interaction? It is not characterized. If it is due to ATP depletion, there is ample precedent for this in the literature.

Fig 5 - what cell line is used for the data presented? Not stated in the figure or legend.

Reviewer #2 (Remarks to the Author):

The manuscript entitled "Targeted production of reactive oxygen species in mitochondria to overcome cancer drug resistance" by Wang et al. reports on an in-vivo and in-vitro study of lipid membrane-coated silica-carbon (LSC) hybrid nanoparticles combined with near-infrared (NIR) to counter the growth of multi-drug resistant tumors.

The authors have shown that using LSC nanoparticles combined with NIR laser irradiation can produce reactive oxygen species (ROS) which can oxidize NADH to NAD⁺. They have also reported that this oxidation can reduce the ATP (Adenosine triphosphate) available for the efflux pumps such as ABC transporters. Further, they show that using this method can make the multi-drug resistant cancer cells lose their resistance for at least five days, giving an opportunity for the chemotherapy.

This manuscript presents new and intriguing results on cellular processes behind the multi-drug resistance and countering it using nanoparticles, which is an important modality in the fields of cancer therapy. Therefore, I strongly think that the paper deserves publication in Nature Communications.

Having said the above, I recommend the following minor corrections , which may improve the paper -

- 1) Line 51: "in the field of oncology in the past several decades." can be changed to "in the field of oncology for the past several decades."
- 2) Line 104 -105: Further insight/explanation could be added to elucidate the kind of chemical reaction that is causing the nanoparticles to lose size after 6 hr, followed by some increase in size after 12 h when exposed to TEOS. Addition of reason for this observed phenomenon would be helpful for future readers.
- 3) Lines 106-107: There is an indication that the pyruvate groups in the nanoparticles decreased due to addition of silica. Does this have any impact on the nanoparticles targeting the mitochondria?
- 4) Lines 120-122 and Fig. 2f, show that the LSC particles have excellent photothermal effect. The increase in temperature is lesser for nanoparticle concentration of 0.25 mg/ml when compared to 0.12 mg/ml. Please explain the reason for this observation for better clarity.
- 5) Line 168: NDA+ should be changed to NAD+.
- 6) Line 199: the line "The production of ROS can be detected at 3 h after NIR laser irradiation, and further increases with time (Fig. 3c)" can be modified to "The production of ROS can be detected at 3 h after NIR laser irradiation, which further increases with time (Fig. 3c)."
- 7) Lines 239-241: in Fig. 3f, there is a decrease in ATP consumption with increase in LSC nanoparticles. Could you please explain why this is observed? Similarly, the authors may explain the reason for decrease in cell viability with increase in nanoparticle concentration in Figs. 5f and 5g.

Point-by-point responses to reviewers

We would like to thank all the reviewers for their insightful and thoughtful comments! We have revised the manuscript according to their advices, which should significantly improve the clarity and quality of our work. Below is a list of the point-by-point responses to the reviewer comments and the corresponding changes that we made. We also highlighted all the changes that we made in response to the reviewers comments in the main text of the manuscript.

Reviewer #1 (Remarks to the Author):

This article reports on lipid membrane-coated silica-carbon (LSC) hybrid nanoparticles that can specifically produce reactive oxygen species (ROS) in mitochondria under near-infrared (NIR) laser irradiation. The intention of these was to treat MDR cancer. The challenge for this manuscript is that the activity of the nanoparticles are empiric - the mitochondrial targeting was not a matter of design, but once observed its mechanism not characterized. The literature is awash with nanoparticle formulations of doxorubicin that are active against NCI-ADR/RES cells and other MDR cell lines, and when the focus shifts to ROS generation, the underlying cause of ROS generation is not determined, it is only characterized. The observational nature of this study leads this reviewer to conclude it is not appropriate for publication in Nature Communications, and is probably more appropriate for a cancer journal.

Re: We thank the reviewer for the critical comments, which definitely helps us to further improve the clarity and quality of our manuscript! We are sorry that we did not present our work clearly in the previous version of this manuscript, which caused the confusion that the activity of our nanoparticle is empiric. In this work, we carefully designed our nanoparticles to compromise the important transmembrane efflux-pump mediated drug resistance mechanism of multidrug resistant cancer cells. We aim to compromise the efflux pump by minimizing the production of adenosine triphosphate (ATP) that drives the efflux pump. Therefore, first, we need to target the ATP factory, the mitochondria, in the multidrug resistant cancer cells. Colloidal carbon sphere (CCS) is an important functional nanoparticle because of the rich functional groups on its surface, which could be synthesized by hydrothermally annealing glucose. Pyruvaldehyde group is one of the functional groups, which is similar to pyruvate that can specifically bind with the monocarboxylate transporters (MCTs) on the outer surface of mitochondria, for possible mitochondria targeting. Following this thought, we first determine the existence of pyruvate in CCS by a specific detection assay (as shown in Supplementary Fig. 1). In order to utilize the enhanced permeability and retention (EPR) of tumor vasculature for in vivo tumor targeting with nanoparticles of ~20-100 nm, the CCS of 200-300 nm was reacted with TEOS to form the silica-CCS (SC) hybrid nanoparticles, which was further coated with lipid membrane to improve their its biocompatibility and form the final LSC nanoparticle. The LSC nanoparticles are ~45 nm, which is excellent for in vivo tumor targeting as shown in Fig. 7a in this revision. As shown in Supplementary Fig. 1, the pyruvate groups are also detectable on the SC and LSC nanoparticles for mitochondria targeting. All these experiments were carefully designed to fulfill the goal of targeting mitochondria together with in vivo tumor targeting and high biocompatibility based on the unique properties of the materials used. The aforementioned information is now incorporated in Abstract, the last 11 lines in the 1st paragraph on page 3, and the last paragraph on page 18 of this revision.

To minimize ATP production in the cells so that minimal energy is available to drive the efflux pump, we desire to interfere the ATP production process by oxidizing NADH (the important intermediate of ATP) into NAD⁺ with reactive oxygen species (ROS) in mitochondria. This is another reason that we started with the CCS to design our nanoparticles, because the surface of the CCS are enriched with both sp² and sp³ hybridized carbon atoms that could catalyze oxidation to produce ROS (i.e., hydrogen peroxide or hydroxyl radicals) according to references 26-28. During NIR laser irradiation, shock

photoacoustic waves could be produced to activate the carbon-steam chemical reactions on the surface of CCS according to references 29-30. At the same time, the increase in temperature during NIR laser irradiation could enhance the catalytic capacity of the sp² and sp³ carbon atoms on the CCS. Therefore, we reason that free radicals could be produced with our LSC nanoparticles when irradiated with NIR laser. Indeed, as shown in Fig. 2e and Supplementary Figs. 8-9 of this revision, the LSC nanoparticles (without any drug) could produce free radicals or ROS under NIR laser irradiation. The aforementioned information is now incorporated in lines 10-17 on page 6 of this revision.

As per the reviewer's advice to further understand the mechanism of ROS production, we conducted experiments to determine the type of free radicals produced by the LSC nanoparticles under NIR laser irradiation. First, singlet oxygen (¹O₂) sensor was used to specifically detect the ¹O₂ free radical. As shown in Fig. 2e and Supplementary Fig. 8, the LSC nanoparticles exhibit an excellent photodynamic capability as demonstrated by the time- and concentration-dependent production of ¹O₂ free radicals in deionized water. Furthermore, 5-(diethoxyphosphoryl)-5-methyl-1-pyrroline-N-oxide (DEPMPO, a spin trap for hydroxyl radicals (·OH)) was used to study free electrons in the systems using the electron paramagnetic resonance spectroscopy. As shown in Supplementary Fig. 9a, there is no obvious peak for the DEPMPO, DEPMPO with NIR laser irradiation (DEPMPO+L), and DEPMPO mixed with LSC nanoparticles (DEPMPO+LSC). In contrast, clear signals can be detected after NIR laser irradiation of the DEPMPO+LSC (i.e., DEPMPO+LSC+L), indicating the ·OH free radicals can be produced by the LSC nanoparticles under NIR laser irradiation. This is further confirmed by the terephthalic acid (TA) assay. TA can react with ·OH to produce 2-hydroxyterephthalic acid (2-HTA) that has a fluorescence peak at ~432 nm. Indeed, the TA solution with LSC nanoparticles after irradiated with NIR laser has stronger fluorescence than all the other control TA solutions. Therefore, two different types of free radicals or ROS can be generated by the LSC nanoparticles under NIR laser irradiation. The aforementioned information is now incorporated in the last paragraph on page 5 and first 10 lines on page 6 of this revision.

We agree with the reviewer that doxorubicin (DOX) with red fluorescence has been widely used as a model drug in the literature for anticancer studies with nanoparticles, although such study has not been reported for NCI/RES-ADR spheres enriched with cancer stem cells (CSCs). Therefore, we conducted more experiments using two more chemotherapy drugs (paclitaxel or PTX in short, and irinotecan or CPT-11 in short) to confirm that the multidrug resistance of the NCI/RES-ADR cancer cells can be overcome with our LSC nanoparticles under NIR laser irradiation. As shown in Fig. 6a-b, the PTX or CPT-11 laden LSC nanoparticles (LSC-P or LSC-C) after irradiated with NIR laser could induce higher cytotoxicity than the free drugs or drug laden nanoparticles alone, to both 2D cultured NCI/RES-ADR cells and 3D cultured NCI/RES-ADR spheres. To further confirm this is due to the suppression of drug resistance, both 2D cultured NCI/RES-ADR cells and 3D cultured NCI/RES-ADR spheres were incubated with empty LSC nanoparticles. After irradiated with NIR laser, the cells were further cultured with free PTX and CPT-11 for 24 h. As shown in Fig. 6c and Supplementary Fig. 41, the cytotoxicity of free PTX and CPT-11 to the 2D and 3D cultured cells with the pre-treatment of LSC nanoparticles and NIR laser irradiation exhibit a dose-dependent manner and is significantly higher than that of the two free drugs to the cells without the pre-treatment. This suggests the drug resistance capability of the cells is suppressed by the LSC+L treatment and free drugs can enter the cells. Indeed, the cellular uptake data (Fig. 6d-e) show that more free PTX (labeled with fluorescein isothiocyanate or FITC in short) and CPT-11 (with blue fluorescence) can enter the 2D and 3D cultured cells treated by the LSC nanoparticles with NIR laser irradiation (LSC+L, +C and LSC+L, +PF), although the highest drug fluorescence can be seen in the cells treated with the encapsulated drugs (i.e., LSC-PF and LSC-C). The latter indicates the advantage of using nanoparticles to for drug delivery. These data with the two additional drugs (PTX and CPT-11) further confirm the drug resistance of the NCI/RES-ADR cells can be overcome with the treatment of LSC nanoparticles and NIR laser irradiation, to enhance the antitumor capacity of the chemotherapy drugs. The aforementioned information is now incorporated in the paragraph across pages

15-16 of this revision.

With the aforementioned clarifications together with new experiments using two more chemotherapy drugs and new experiments on the mechanisms of mitochondria targeting and ROS production, we sincerely hope that the reviewer agree that our nanoparticles and experiments are carefully designed to overcome the mechanism of cancer multidrug resistance and this work is appropriate for publication in *Nature Communications*.

As an example of utility, the authors encapsulated doxorubicin into the LSC nanoparticles. The manuscript then moves on to mitochondrial targeting. The authors argue that the nanoparticles can target mitochondria ‘die to the pyruvate’, and the localization is confirmed using TEM. While the localization is demonstrated, the mechanism is speculated, but not characterized. Given the negative potential at the mitochondrial membrane, a net positive charge on a small molecule is usually associated with mitochondrial accumulation - the mechanism for mitochondrial targeting would need to be elucidated to sufficiently strengthen this manuscript.

Re: We are sorry for the confusion! This is probably due to the misleading sketch in Fig. 1 of the previous version of this manuscript, where the positively charged DOX is shown to accumulate in mitochondria by itself. This mistake is now corrected in this revision, and the mechanism of mitochondrial targeting is better clarified/elucidated by redrawing the sketch in Fig. 1 and by conducting more experiments according to the reviewer’s advice. In this study, we designed a novel LSC nanoparticles that have a negatively charged surface as shown in Supplementary Fig. 24, to target mitochondria. Our data on intracellular distribution of free DOX show that free DOX does not target mitochondria by itself although it is a positively charged small molecule. As shown in Supplementary Figs. 12-14, free DOX does not enter the multidrug resistant NCI/RES-ADR cancer cells and it is mainly located in the nuclei of the non-drug resistant MCF-7 and OVCAR-8 cancer cells. We also conduct experiments to take the transmission electron microscopy (TEM) images of the NCI/RES-ADR cells treated with free DOX. As shown in Supplementary Fig. 22, there is no clear difference in the cellular structure between the free DOX treated cells and the cells without any drug treatment (Fig. 3b). All these results suggest that free DOX does not target mitochondria by itself. The aforementioned information is now incorporated in the last 13 lines in the 1st paragraph on page 3, and the 1st paragraph on page 10 of this revision.

According to the reviewer’s advice, we conducted more experiment to characterize and elucidate the mechanism of mitochondria targeting with our LSC nanoparticles. First, we prepared a lipid coated silica nanoparticle (LS) using same procedure for preparing LSC nanoparticles except that no colloidal carbon sphere (CCS) was used. As shown in Supplementary Fig. 20a, the distribution of the DOX laden-LS nanoparticles (LS-D) is different from the distribution of mitochondria, suggesting the LS nanoparticles without the colloidal carbon do not target mitochondria. To further confirm this, the intracellular distribution of LS-D nanoparticles was checked with TEM. As shown in Supplementary Fig. 21a, none of the LS nanoparticles is located in mitochondria. These results indicate that the crucial role of the CCS in rendering the LCS nanoparticles with the important property of mitochondria targeting. To further confirm the pyruvate-mediated targeting of mitochondria with the LCS nanoparticles, we conducted more experiments to pre-treated/blocked the NCI/RES-ADR cells with pyruvic acid for 6 h before incubating them with the LSC nanoparticles. Indeed, the distribution of LSC-D nanoparticles is no longer similar to that of mitochondria according to the confocal fluorescence images (Supplementary Fig. 20b). The TEM images also show that pre-treating the cells with pyruvic acid minimizes mitochondria targeting with the LSC nanoparticles (Supplementary Fig. 21b). We further calculated the percentage of the endosome/lysosome-escaped LSC (with or without pre-blocking using pyruvic acid) or LS nanoparticles (without pre-blocking using pyruvic acid) within mitochondria. As shown in Supplementary Fig. 21c, more than 40% of the endosome/lysosome-escaped LSC-D nanoparticles are within mitochondria while it

is 0% for the LS-D nanoparticles. With pre-blocking using pyruvic acid, the percentage decreases from more than 40% to ~3%. Taken together, these data support that the pyruvate group on the surface of the LSC nanoparticles is responsible for their capability of targeting mitochondria. The aforementioned information is now incorporated in the first 19 lines on page 9 of this revision.

It is impossible to conclude from Fig 3B that the nanoparticles are targeting the mitochondria - there appear to be significant numbers of nanoparticles outside the mitochondria all - can the particles be counted and quantified across multiple fields of view? There appears to be an absence of free doxorubicin controls in Figure 3.

Re: We agree that there were significant numbers of LSC-D nanoparticles outside the mitochondria in the TEM image according to Fig. 3b, and the nanoparticles outside mitochondria are mainly located in the endo/lysosomes through which the nanoparticles are taken up by cells via endocytosis (see Supplementary Figs. 12-14). As per reviewer's advice, we calculated the percentage of the LSC-D nanoparticles in mitochondria out of all the LSC-D nanoparticles escaped out of the endo/lysosomes in the cells. As shown in Supplementary Fig. 21c, more than 40% of the LSC-D nanoparticles escaped from the endo/lysosomes are in mitochondria. To further support the mitochondria targeting capability of the LSC nanoparticles, we conducted more experiments (1) to synthesize nanoparticles using lipid and silica (LS nanoparticles) without CCS to study the intracellular distribution of the DOX laden LS (LS-D) nanoparticles, and (2) to investigate the effect of pre-blocking the cells with pyruvic acid on the mitochondria targeting capability of the LSC nanoparticles. As also shown in Supplementary Fig. 21a and c, 0% of the LS-D nanoparticles escaped from the endo/lysosomes are observed in mitochondria. With pre-blocking the cells using pyruvic acid, the percentage of LSC-D nanoparticles escaped from the endo/lysosomes in mitochondria decreases to ~3% (Supplementary Fig. 21b-c). These data suggest that the LSC nanoparticles may target mitochondria via the pyruvate group on their surface. It is worth noting that to prepare cells for TEM imaging, thin slices of ~50 nm were cut through the cells. Considering the mitochondria are ~0.75-3 μm in diameter (reference 40), the TEM images only show ~1/15-1/60 of the whole mitochondria. As a result, only few LSC-D nanoparticles are observable in mitochondria on the TEM images and the actual number of LSC-D nanoparticles in mitochondria could be ~15-60 times of that observed in the TEM images. The aforementioned information is now incorporated in the paragraph on page 9 of this revision.

According to reviewer's advice, more experiments were conducted for the free DOX controls in Fig. 3 and the data are shown Supplementary Figs. 22 and 26. There is no clear difference in the cellular structure between the free DOX treated cells and the cells without any drug treatment (Supplementary Fig. 22). This is probably because the NCI/RES-ADR cells are multidrug resistant so that free DOX cannot enter the cells to have any impact on the cells (Supplementary Fig. 12). Moreover, the ROS production by the free DOX or NIR laser irradiation treatment alone is minimal (Supplementary Fig. 26), compared with the treatment of LSC nanoparticles with NIR laser irradiation (LSC+L). The aforementioned information is now incorporated in the 1st paragraph on page 10 and the last 3 lines in the 1st paragraph on page 11 of this revision.

Again, it is observed that (presumably) P-gp is not active in cells treated with nanoparticles (yet another mechanism of action for MDR sensitization?) - is this due to the ROS generation? Or a specific interaction? It is not characterized. If it is due to ATP depletion, there is ample precedent for this in the literature.

Re: Yes, the dysfunction of the P-gp efflux pumps is due to both ROS production and specific interactions. We designed the LSC nanoparticles to specifically target mitochondria via the pyruvate group on the surface of the nanoparticles. Under NIR laser irradiation, ROS could be specifically produced by the nanoparticles in mitochondria and react with NADH to interfere the production of ATP

in mitochondria, which cut the energy supply to the drug efflux pumps. This strategy of compromising the function of the efflux pump by specifically generating ROS in mitochondria via pyruvate-mediated targeting to oxidize NADH and minimize ATP synthesis, has never been reported in the literature. The increased generation of ROS and the decreased amount of NADH in the cells treated with the LSC nanoparticles and NIR laser irradiation (LSC+L) are characterized and shown in Fig. 3d and 3e, respectively. As a result, the P-gp efflux pumps in the cells with the LSC+L treatment lack ATP, which is confirmed by their high consumption of ATP after isolation of the efflux pumps from the cells (Fig. 3f). Consequently, The P-gp efflux pumps in the cells with the LSC+L treatment become dysfunctional, which allows free drugs to enter both 2D cultured NCI/RES-ADR multidrug resistant cells and 3D cultured NCI/RES-ADR spheres as shown in Fig. 6d-e and Supplementary Figs. 32 and 33. As per the reviewer's advice, we conducted more experiments to investigate other potential mechanisms that may sensitize the multidrug resistant NCI/RES-ADR cells to chemotherapy drugs besides minimizing ATP production. As shown in Fig. 4a, most of the P-gp efflux pumps are located on the cell plasma membrane for control cells without any treatment. Interestingly, for LSC+L treated cells, many of the P-gp efflux pumps are distributed in the cytoplasm. This might be due to the minimized ATP production so that there is no enough energy supply to transport the P-gp efflux pumps to the cell plasma membrane. To support this, we incubated the multidrug resistant cells with oligomycin (an inhibitor of ATP synthase) for 12 h to check the distribution of the P-gp efflux pumps. As shown in Fig. 4a, the P-gp efflux pumps also distributed in both the cell plasma membrane and cytoplasm, similar to the LSC+L treated cells. These results suggest the ATP not only provide the energy for the P-gp efflux pump to function, but also is needed to transport the P-gp efflux pumps to the cell plasma membrane. The total amount of P-gp in the multidrug resistant cells from the aforementioned three groups was studied using flow cytometry. As shown in Fig. 4b-c and Supplementary Fig. 34, the expression of P-gp is slightly but significantly decreased compared with control and oligomycin-treated cells. This together with the increased intracellular distribution of the P-gp should decrease the amount of the P-gp efflux pumps on the plasma membrane of the LSC+L treated cells, which should also reduce the drug resistant capacity of the multidrug resistant cells. The aforementioned information is now incorporated in the 2nd paragraph on page 13 of this revision.

We further conducted experiments to check how the LSC+L treatment could affect the expression of some stress-related proteins in the multidrug resistant NCI/RES-ADR cells. As shown in Supplementary Fig. 45a-b, the expression of the 70-kDa heat shock protein (HSP70) is significantly decreased at 12 h after the LSC+L treatment compared with control, NIR laser, or LSC nanoparticles treated cells although no significant change is observable for the expression of 90-kDa heat shock protein (HSP90). HSP70 is abnormally overexpressed in majority of cancer cells and important for cancer cell survival (reference 42). Therefore, the decrease in HSP70 could make the cancer cells more sensitive to chemotherapy drugs (reference 43). Moreover, the expression of heat shock factor-1 (HSF-1) that is commonly thought to be associated with HSP70 (reference 44), is also decreased at 12 h after the LSC+L treatment. Previous studies have shown that the high HSF-1 expression is associated with poor outcome of chemotherapy (including lung, breast, and colon cancer) and increases cancer stem cell (CSC) frequency (references 45 and 46). Therefore, the decreased expression of HSF-1 is beneficial for chemotherapy of not only regular cancer cells but also the CSCs. It is also worth noting that the HSP70 and HSF-1 protein complex is required for the stabilization of mutant p53 protein in cancer cells (reference 47). The decrease of both HSP70 and HSF-1 should compromise the stability of the mutant p53 protein. Indeed, the mutant p53 protein is highly expressed in the NCI/RES-ADR cells and the LSC nanoparticles or NIR laser irradiation alone has no impact on the expression of the mutant p53 in the multidrug resistant cells. Importantly, it is significantly decreased in the LSC+L treated cells. Similarly, the mutant p53 is important to the resistance of cancer cells to both apoptosis and chemotherapy (reference 48). We further confirmed the aforementioned observations from western blotting studies using flow cytometry (Supplementary Fig. 46). Confocal fluorescence images show that the HSF-1 is distributed in both the cytoplasm and nuclei of

cells in the control, NIR laser, or LSC groups, but decreased in the LSC+L group (Supplementary Fig. 47a). HSP70 mainly distributes in the cytoplasm while mutant p53 proteins are mainly observed in nuclei of cells for the control, laser, or LSC groups, and both are decreased in LSC+L treated cells (Supplementary Fig. 47b-c). Previous studies show that the expression of P-gp and mutant p53 shares the same trend (i.e., both are high or low, instead of one being high while the other is low) in clinical samples (reference 49). In addition, both HSP70 and HSF-1 have been shown to regulate the expression of P-gp in cancer cells (references 50 and 51). Therefore, in this study, the decreased expression of P-gp is possibly due to the decreased expression of HSP70, HSF-1, and mutant p53. The detailed signaling pathways that regulate the expression of P-gp is beyond the scope of this study, and warrants further investigation in future studies. The aforementioned information is now incorporated in the paragraph across pages 20-21 of this revision.

In summary, targeted production of ROS in mitochondria under NIR laser irradiation of LSC nanoparticles can minimize the formation of ATP, which further compromises the P-gp activity on the cell plasma membrane as drug efflux pumps. Under NIR laser irradiation, the LSC nanoparticles not only cut off the energy supply to the P-gp efflux pumps, but also affect its expression and cellular distribution. In addition, decreasing the expression of HSP70, HSF-1, and mutant p53 could render cancer cells more sensitive to chemotherapy. Collectively, multiple mechanisms have been identified to overcome the drug resistance capability of the multidrug resistant NCI/RES-ADR cancer cells and improve their sensitivity to chemotherapy drugs.

Fig 5 - what cell line is used for the data presented? Not stated in the figure or legend.

Re: For the *in vivo* data shown in the previous Fig. 5 (Fig. 7 in this revision), detached NCI/RES-ADR sphere cells were used to obtain NCI/RES-ADR xenograft. This is now clarified in the caption of Fig. 7 of this revision. Again, we thank the reviewer for all the insightful and thoughtful comments!

Reviewer #2 (Remarks to the Author):

The manuscript entitled "Targeted production of reactive oxygen species in mitochondria to overcome cancer drug resistance" by Wang et al. reports on an in-vivo and in-vitro study of lipid membrane-coated silica-carbon (LSC) hybrid nanoparticles combined with near-infrared (NIR) to counter the growth of multi-drug resistant tumors.

The authors have shown that using LSC nanoparticles combined with NIR laser irradiation can produce reactive oxygen species (ROS) which can oxidize NADH to NAD⁺. They have also reported that this oxidation can reduce the ATP (Adenosine triphosphate) available for the efflux pumps such as ABC transporters. Further, they show that using this method can make the multi-drug resistant cancer cells lose their resistance for at least five days, giving an opportunity for the chemotherapy.

This manuscript presents new and intriguing results on cellular processes behind the multi-drug resistance and countering it using nanoparticles, which is an important modality in the fields of cancer therapy. Therefore, I strongly think that the paper deserves publication in Nature Communications.

Having said the above, I recommend the following minor corrections, which may improve the paper –

Re: We thank the reviewer for the insightful and thoughtful comments! All the concerns are now fully addressed in this revision as detailed below.

1) Line 51: "in the field of oncology in the past several decades." can be changed to "in the field of

oncology for the past several decades.”

Re: Thank you for the advice! The change is now made in line 5 in the 2nd paragraph on page 2 of this revision.

2) Line 104 -105: Further insight/explanation could be added to elucidate the kind of chemical reaction that is causing the nanoparticles to lose size after 6 hr, followed by some increase in size after 12 h when exposed to TEOS. Addition of reason for this observed phenomenon would be helpful for future readers.

Re: We agree! The colloidal carbon sphere (CCS) is a porous material, and when it is exposed to TEOS, the TEOS may enter the porous space and react with CCS to form new chemical bonds (Si-O-Si between TEOS and Si-O-C between TEOS and CCS, Supplementary Figs. 4 and 5) under the experimental condition of this study. Therefore, there might be two events that may affect the size of the nanoparticles: (1) the formation of the new chemical bonds may generate some cohesive forces to pull the molecules in CCS closer to decrease the size of the nanoparticles, and (2) the addition of silica in the porous space and on the surface of the CCS to increase the size of the nanoparticles. It is possible that the former event dominates the latter during the first 6 h while the latter is more important than the former from 6 to 12 h, in determining the size of the nanoparticles during the process. The aforementioned information is now incorporated in the 1st paragraph on page 19 of this revision.

3) Lines 106-107: There is an indication that the pyruvate groups in the nanoparticles decreased due to addition of silica. Does this have any impact on the nanoparticles targeting the mitochondria?

Re: We are sorry for the confusion! The amount of pyruvate groups in the various nanoparticles shown in Supplementary Fig. 1 is based on the same weight (50 μ g) rather than the same number of the various nanoparticles. Given the same weight, the CCS content (100% for the CCS nanoparticles) in the samples of SC (-3h and -6h) and LSC nanoparticles should decrease because they also contain silica (for SC nanoparticles) and both silica and lipid (for LSC nanoparticles). This results in the decrease in the amount of the pyruvate groups in the 50 μ g of SC and LSC nanoparticles. In fact, each SC or LSC nanoparticle is produced from one CCS nanoparticle, as schematically illustrated in Fig. 2a. Therefore, the total amount of pyruvate groups should be similar in each of the SC, LSC, and CCS nanoparticles. In view of this, the density of pyruvate groups on the surface of the SC and LSC nanoparticles is expected to be even higher than that on the CCS nanoparticles. This is because the size and surface area of the SC (~35 nm in diameter on average) nanoparticles are decreased by ~ 7 ($=250/35$) and ~ 51 ($=(250/35)^2$) times on average, respectively, compared to the CCS nanoparticles (~250 nm in diameter on average). Since LSC nanoparticles were made by coating lipid on the CS nanoparticles via its interaction with APTMS that is a silane-coupling agent (i.e., interacts with silica), the LSC and SC nanoparticles are expected to have similar density of pyruvate groups on their surface. Therefore, the addition of silica and lipid in this study should not greatly affect (and may even improve) the capability of the nanoparticles in targeting mitochondria. The aforementioned information is now incorporated in the 2nd paragraph on page 19 of this revision.

4) Lines 120-122 and Fig. 2f, show that the LSC particles have excellent photothermal effect. The increase in temperature is lesser for nanoparticle concentration of 0.25 mg/ml when compared to 0.12 mg/ml. Please explain the reason for this observation for better clarity.

Re: We are sorry for the mix-up! It is now corrected in Fig. 2f of this revision.

5) Line 168: NDA+ should be changed to NAD+.

Re: Thank you! The change is made as per the advice in line 7 in the 2nd paragraph on page 8 of this revision.

6) Line 199: the line “The production of ROS can be detected at 3 h after NIR laser irradiation, and further increases with time (Fig. 3c)” can be modified to “The production of ROS can be detected at 3 h after NIR laser irradiation, which further increases with time (Fig. 3c).”

Re: Thank you! The change is made as per the advice in the last line on page 10 of this revision.

7) Lines 239-241: in Fig. 3f, there is a decrease in ATP consumption with increase in LSC nanoparticles. Could you please explain why this is observed? Similarly, the authors may explain the reason for decrease in cell viability with increase in nanoparticle concentration in Figs. 5f and 5g.

Re: In Fig. 3f, the decrease in ATP consumption is significant for the condition with a LSC nanoparticle concentration of 1.6 mg/ml. This should be due to the cytotoxicity of the treatment with the high concentration of LSC nanoparticles and laser irradiation (see Fig. 5c in this revision), because it is more difficult to collect all dead cells than live cells for the measurement of ATP consumption. In other words, the ATP consumption is decreased due to the decrease in cell number for the condition with a LSC nanoparticle concentration of 1.6 mg/ml. The significant cytotoxicity of the LSC nanoparticles at high concentrations (e.g., 1.6 mg/ml) is probably due to the combined photothermal and photodynamic effects of the LSC nanoparticles. The aforementioned information is now incorporated in the last 4 lines in the 1st paragraph on page 14 of this revision. We thank the reviewer again for all the insightful and thoughtful comments!

REVIEWERS' COMMENTS:

Reviewer #1 (Remarks to the Author):

Apologies for the delay in responding, and thank you to the authors for such detailed responses. I have re-read the manuscript and figures in their updated and edited forms, along with the response letter. I greatly appreciate the updated Figure 1, which significantly aids the reader in understanding the article as presented.

The new data addressing the underlying mechanistic mechanisms at play significantly improve the manuscript, particularly demonstrating that the effects of NPs extend beyond doxorubicin to other drugs. I believe the insight into the role nanoparticle formulations (with appropriate design) may play in tackling multidrug resistant cancers make this article appropriate for publication in Nature Communications in its current updated form.

Reviewer #2 (Remarks to the Author):

Critiques has been adequately addressed.